# Habit formation viewed as structural change in the behavioral network

Kota Yamada [1,2 ✉] & Koji Toda [1]

Habit formation is a process in which an action becomes involuntary. While goal-directed behavior is driven by its consequences, habits are elicited by a situation rather than its consequences. Existing theories have proposed that actions are controlled by corresponding two distinct systems. Although canonical theories based on such distinctions are starting to be challenged, there are a few theoretical frameworks that implement goal-directed behavior and habits within a single system. Here, we propose a novel theoretical framework by hypothesizing that behavior is a network composed of several responses. With this framework, we have shown that the transition of goal-directed actions to habits is caused by a change in a single network structure. Furthermore, we confirmed that the proposed network model behaves in a manner consistent with the existing experimental results reported in animal behavioral studies. Our results revealed that habit could be formed under the control of a single system rather than two distinct systems. By capturing the behavior as a single network change, this framework provides a new perspective on studying the structure of the behavior for experimental and theoretical research.

[1] Department of Psychology, Keio University, Tokyo, Japan. [2] Japan Society for Promotion of Science, Tokyo, Japan. ✉email: haroldthebarrel.yk@gmail.com

To behave flexibly in a given environment, organisms need to choose their actions based on the consequences of the actions. This type of behavior is called goal-directed behavior. As we keep repeating the same action under a certain situation, the action is elicited by the situation rather than its consequences. This type of behavior is called a habit. Goal-directed behavior requires high computational resources because organisms must process the information about their external environment and how their actions affect it. In contrast, habit shows a more stereotyped and less flexible behavior, requiring less computation. In this sense, habit formation can be viewed as the optimization process of energy consumption by the organism.

Existing theories about habit formation are based on evidence from experimental or theoretical research in psychology and neuroscience. In the canonical view, responses are controlled by two different systems: goal-directed and habit systems. Such theories proposed that goal-directed and habit systems control responses by assigning different weights, and the difference in the weights determines whether the response is goal-directed or habit[1,2]. In this assumption, habit formation can be viewed as losing control by the consequence of the response or reward sensitivity. However, some models explain habits in a multistage Markov decision task and challenge the canonical dichotomy of goal-directed and habits systems[3,4]. In addition, some researchers reviewed existing studies on habit formation and cast doubt on the canonical framework of habit formation by showing the possibilities that habits are also controlled by their consequences[5,6].

In contrast to the canonical view, Dezfouli and Balleine[7] proposed a new perspective that habit formation can be viewed as shaping or acquiring response sequences. In their model, an agent chooses their goal in a goal-directed manner and generates a response sequence to reach there. Although habits are viewed as a lack of reward sensitivity in the canonical view, their new model considers stereotyped behaviors as acquired response sequences. To what extent could this model change the way of viewing accumulating evidence of habit formation? Garr and Dalamater[8] shows that rats acquired stereotyped response sequences did not lose reward sensitivity. In a series of studies reported by Dezfouli and Balleine[7,9,10] dealt with only a few experiments on the reward sensitivity in free operant situations[11–15]. Another approach employs the planning process[3,4]. Pezzulo et al.[3] stressed the importance of planning in goal-directed behaviors and built a single mixed-controller model consisting of goal-directed behaviors and habits. Keramati et al.[4] proposed that the canonical goal-directed and habits systems can be viewed as edges of the spectrum by building an integrated model of goal-directed planning and habits. Although application of their models was limited to the multistage choice task, the model could serve as a basis for a novel model with common assumptions and additional applicability in experiments on reward sensitivity in free situations[11–15].

Here, instead of assuming two explicitly distinguished mechanisms as in the canonical views, we consider behavior as a network consisting of multiple responses and show that changes in the structure of the network cause two behavioral features, goal-directed behavior and habit. By doing so, we could explain the lack of reward sensitivity in habit formation, which is a characteristic of the canonical view on habits.

**Behavioral network**. There are two methodological approaches for studying animal behavior. One stream is an in-laboratory psychological approach that studies the behavior of animals, including humans, under experimentally controlled environments. Here, investigators measure only experimentally defined responses of subjects (lever press, key peck, nose poke, freezing, salivation, licking, eye blink, etc.) or put them into rigidly controlled situations where they can only engage in the responses to the well-defined stimulus. Another stream is an ethological approach that studies animal behavior under more natural and ecologically valid environments[16]. In this case, behavior that the organism is engaged in the real world could be observed, but the stimulus is difficult to control in terms of the strength, frequency, timing, etc. Although these two approaches seem to conflict with each other, both are complementary for understanding behavior and its biological substrates. Recent advances in machine learning have allowed us to objectively measure the detailed structure of behaviors[16–18]. Animals are engaged in more than lever press, key peck, or nose poke, they approach and orient to the stimulus, and walk or sniff around and explore in the given environment. Although the importance of observation and measurement of the behavior during learning was attempted in classic behavioral studies[19–22], current behavioral quantification methods are expected to reveal the relationship between behavior and its underlying mechanism in a way that integrates the different disciplines of psychology, neuroscience, and ethology[23,24]. However, conventional views on behavior in psychology and neuroscience are based on empirical results obtained from the approaches before the appearance of such a new quantification technique of the behavior. Here, we present a new theoretical novel framework that focuses on how behavior is organized and how its structure brings specific characteristics to behavior.

Existing studies measured only specific experimenter-defined responses of animals including humans, and ignored various responses that the animals actually engaged in. However, there is considerable evidence that animals engage in various responses which affect the learned responses. For example, animals engage in a specific response immediately after the reward presentation[25–27], engage in responses irrelevant to an experiment[20], show a specific response sequence between reward presentations[21], or show a specific response that counteracts learned responses[28]. Theoretically, some characteristics of operant responses are explained by assuming the existence of other responses[29–33]. These experimental facts and theoretical assumptions indicate that animal responses do not exist in isolation but are associated with other responses. We assume such relationships between responses as a network in which responses and transitions between them are considered nodes and edges, respectively.

Network science emerged in the mid to late 1990s and has spread to a wide range of fields. One of the important aspects of network science is handling the structure of the network. For example, in a network in which individual nodes are randomly connected, the distance between each node is large and the information transmission is slow. However, if there is a node called a hub in the network, which has acquired a large number of edges from other nodes, information can be rapidly transmitted through that node. In reality, this is like an influencer sending out information on a social networking service, which attracts the attention of a larger number of users and rapidly spreads the information. In this way, the structure of the network is closely related to the behavior of the entire system. We introduce this perspective of network structure to behavioral science. In this view, each response is assumed as a node, and behavior could be captured as a network of interconnected nodes. By doing so, we try to explain existing behavioral phenomena from a new perspective of the overall structure of behavior. Introducing the concept of network science to experimental analysis of behavior and the theory of habit formation has not been focused on so far.

Here, we provide a computational formulation of the behavioral network and explain habit formation from the

viewpoint of changes in the network structure. In simulation 1, we generated an arbitrary network and examined what kind of structure forms habit, and showed that habit formation occurs when edges are concentrated on a specific response. In Simulation 2, we examined whether the factors reported to promote or inhibit habit formation from existing behavioral studies have similar effects on the proposed model. There are three important factors on habit formation: (1) the amount of training[11,12], (2) the schedule of rewards[13], and (3) the presence or absence of choice[14,15]. The effects of these factors on the proposed model were consistent with the existing experimental results. These results imply that habit formation can be explained not by the control of the two systems, but by a single system constituting the change in the structure of the behavioral network. Furthermore, the results demonstrate that all responses are goal-directed, rather than the conventional dichotomy of goal-directed and habitual behaviors.

## Results

We considered the behavior of an agent as a network consisting of different categories of responses (e.g., lever pressing, grooming, stretching, etc.). Each response was assumed to be a node, and the transition between responses was assumed to be an edge (Fig. 1a). The purpose of our agent was the same as the normal reinforcement learning setting of reward maximization. To achieve it, the agent's behavior was modeled by choices based on the values of rewards and the shortest path from the currently engaging response to the chosen response. Although this modeling differed from the ordinary setting, it accounted for the behavior of organisms in the natural environment. Our model reflected three facts (Fig. 1b). (1) Most organisms, including humans, engage in various responses in their lives. For example, a rat in a free-operant experiment presses a lever in one moment and grooms its hair or explores the experimental apparatus the next moment. (2) The responses are associated with different types of rewards. Lever pressing is associated with food presentation. Hair grooming is associated with removing disconformity. Exploring within the apparatus is associated with escaping from the apparatus. (3) When an animal shifts from the currently engaging response to another response, it may choose to reach the response via relatively fewer responses. For example, if a rat engages in sniffing (Fig. 1b left) and then chooses to press a lever (Fig. 1b center), two paths or response sequences are available: walking to the front of the lever and pressing the lever or walking to the front of the lever followed by grooming and then pressing the lever (Fig. 1b center). Grooming requires additional time and is redundant for pressing the lever. Thus, the rat may choose the shortest path, i.e., walk to the front of the lever and press it (Fig. 1b right). In a large behavioral space, random search increases the time required to reach the desired response and does not warrant reaching the desired response. In summary, the agent chooses one available response associated with different rewards and reaches the chosen response by following the shortest path from the currently engaging response. The agent loops through this process in the behavioral network, which is composed of responses.

We assumed that how nodes in a network and attachment of an edge between two nodes depended on the history of past rewards experienced by the agent. We employed Q-learning[34] to represent the history of rewards obtained when transitioning from one response to another. In ordinary Q-learning, an agent learns the action-value in a state. However, since our model dealt with transitions between responses, we treated the response of the agent as a state. Thus, Q-learning in our model was represented by the following equation, assigning the response a time point prior to the state:

$$Q(a_{t-1}, a_t) \leftarrow Q(a_{t-1}.a_t) + \alpha \cdot \delta \qquad (1)$$

In this equation, $\alpha$ denotes the learning rate, we set $\alpha = 0.1$ for all simulations; and $\delta$ is the reward prediction error (or temporal difference error). The reward prediction error was calculated as follows:

$$\delta = R(a_t) + \gamma \cdot \max_{a_{t+1}} Q(a_t, a_{t+1}) - Q(a_{t-1}, a_t) \qquad (2)$$

In this equation, $\gamma$ denotes the discount rate of future rewards and we set $\gamma = 0.5$ for all simulations. $R_t$ denotes the reward obtained by a transition, and the reward functions are different between simulations, which have been explained in detail in "Materials and Methods" section.

The probability that an edge is attached between any two nodes depends on the Q-value and is calculated using the softmax function. The probability was calculated using the following equation:

$$p_{i,j} = \frac{e^{-\beta_n Q(i,j)}}{\sum_{j=1}^{N} e^{-\beta_n Q(i,j)}} \qquad (3)$$

In this equation, $N$ denotes the number of nodes in the network and all the responses that the agent can engage in. $\beta_n$ denotes the inverse temperature and we set $\beta_n = 50$ in all simulations. We also sampled two edges according to Eq. (3), such that every node had at least two edges. We used "networkx," a Python library for network analysis, to generate the network.

The algorithm for the agent to choose a response contains two steps: (1) choice of the response based on the value of the reward, and (2) searching the shortest path from the current engaging response to the chosen node. In the choice of the response based on the value of the reward, the probability of choosing a response is calculated by proportional allocation of the reward value. The shortest path search includes selecting the shortest path between the current response to the chosen response and the agent engaging in the responses containing the path in sequence.

The probability of response $i$ was calculated according to the following equation:

$$p_i = \frac{r_i}{\sum_{j=1}^{N} r_j} \qquad (4)$$

In this equation, $r_i$ denotes the value of the reward obtained from response $i$. In our simulation, the value of the reward obtained from the operant response was 1.0, and the other response was 0.001.

The shortest path search is used to find the shortest path between any two nodes in the network. We employed Dijkstra's algorithm[35] in all our simulations. If there were multiple shortest paths between any two nodes, we randomly choose one of them. We implemented the path search by using NetworkX[36].

**Simulation 1: network structure and habit formation**. In the Simulation 1, we searched for the structure of the network where habits formation occurs. First, we generate a network based on the Q-matrix. We used an arbitrary Q-matrix to operate the degree of the edge concentration on the operant response. The Q-matrix is defined as the direct product of the Q-vector. The Q-vector contains scalars ranging from 0. to 1. and each element corresponds to each response. More specifically, The first element corresponds to the operant response and others correspond to the other responses. In simulation 1, we fixed the value for the other responses to 0.001 and varied the value for the operant response, Q-operant, from 0.0 to 1.0. To examine the degree of habit, we used the reward devaluation procedure used in free-operant

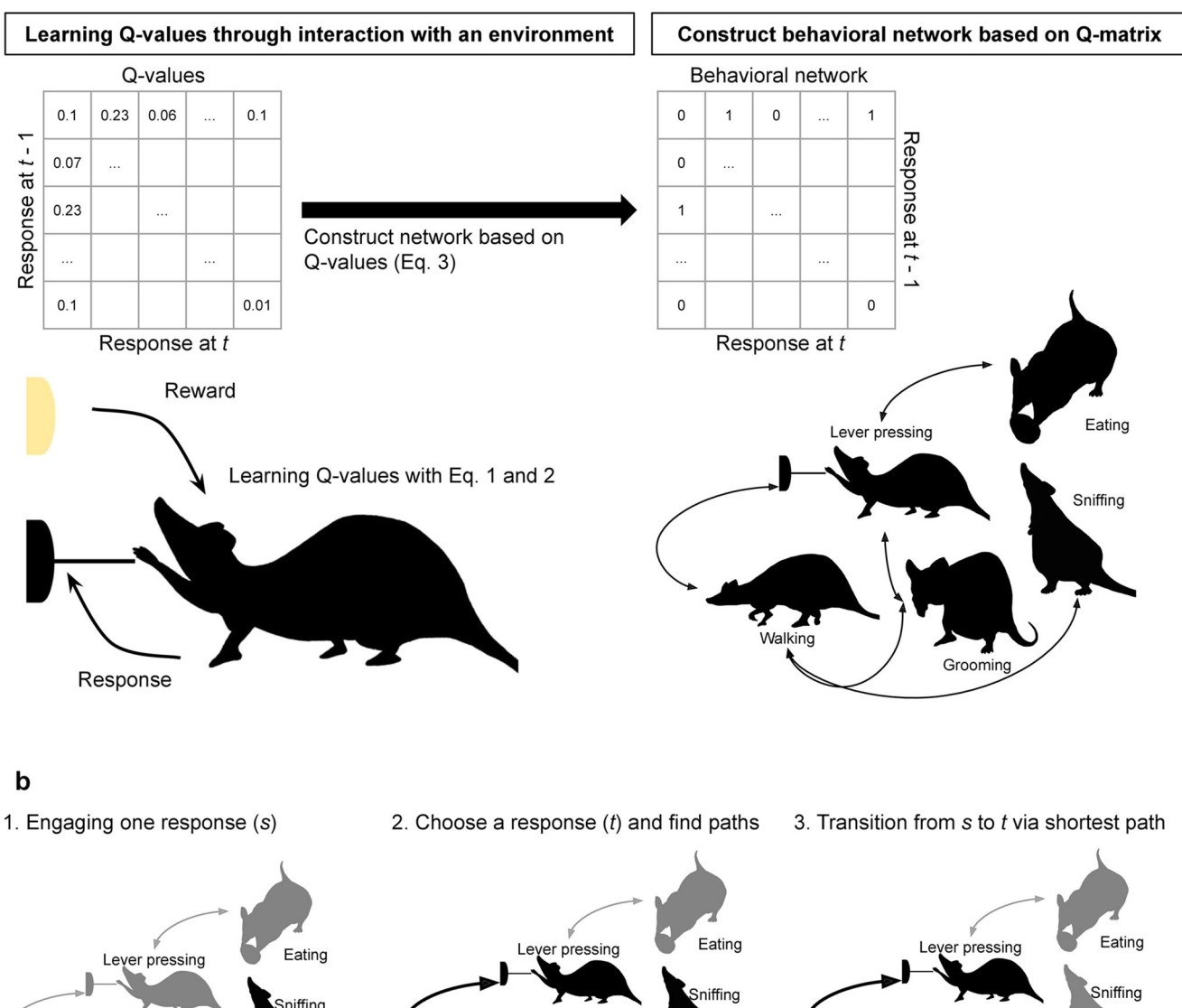

**Fig. 1 Scheme of the behavioral network. a** The schematic representation of the behavioral network model represents how agents learn the Q-values by interacting with the environment and generate a behavioral network based on these values. The behavioral network consists of multiple responses. **b** The schematic representation of the model's behavior shows how the agents transit in the network. The left panel shows the initial state in which agents engage in a response. The center panel shows that agents choose a goal and search for the shortest path. The right panel shows that agents transit from the initial response to the goal via the shortest path.

experimental situations. The earliest demonstrations of habit formation by Dickinson et al.[11–13] used the reward devaluation procedure. In this procedure, the investigators train the animals to press the lever with a reward. After the animal learned lever pressings to obtain the reward, the value of reward was reduced by poisoning it with lithium chloride. In this procedure, animals learnt the reward value outside the experiment. Subsequently, investigators examined if the animal pressed the lever without reward deliveries, or an extinction test. Thus, the reward value for the animal was not updated in the test. When the animal pressed the lever, the reward was poisonous, and the responses were considered to be a habit. When the lever-presses decreased after devaluation, the responses were considered to be goal-directed

behavior. To reproduce the procedure in the simulation setting, we set up the baseline and post devaluation phases where the value of reward obtained by the operant response is 1 and 0, respectively. As animals had experienced reward devaluation outside the experiments in the experimental setting, our agents did not update the reward value within the simulation but changed it from 1.0 to 0.0 before starting when moving from baseline to post devaluation phases. In both baseline and post devaluation phases, the first response that the agent engaged was randomly determined. Then, the agent chooses a response based on the reward value and searches for the shortest path to the response from the current engaging response. They engage in responses contained in the path and the agent reaches the chosen

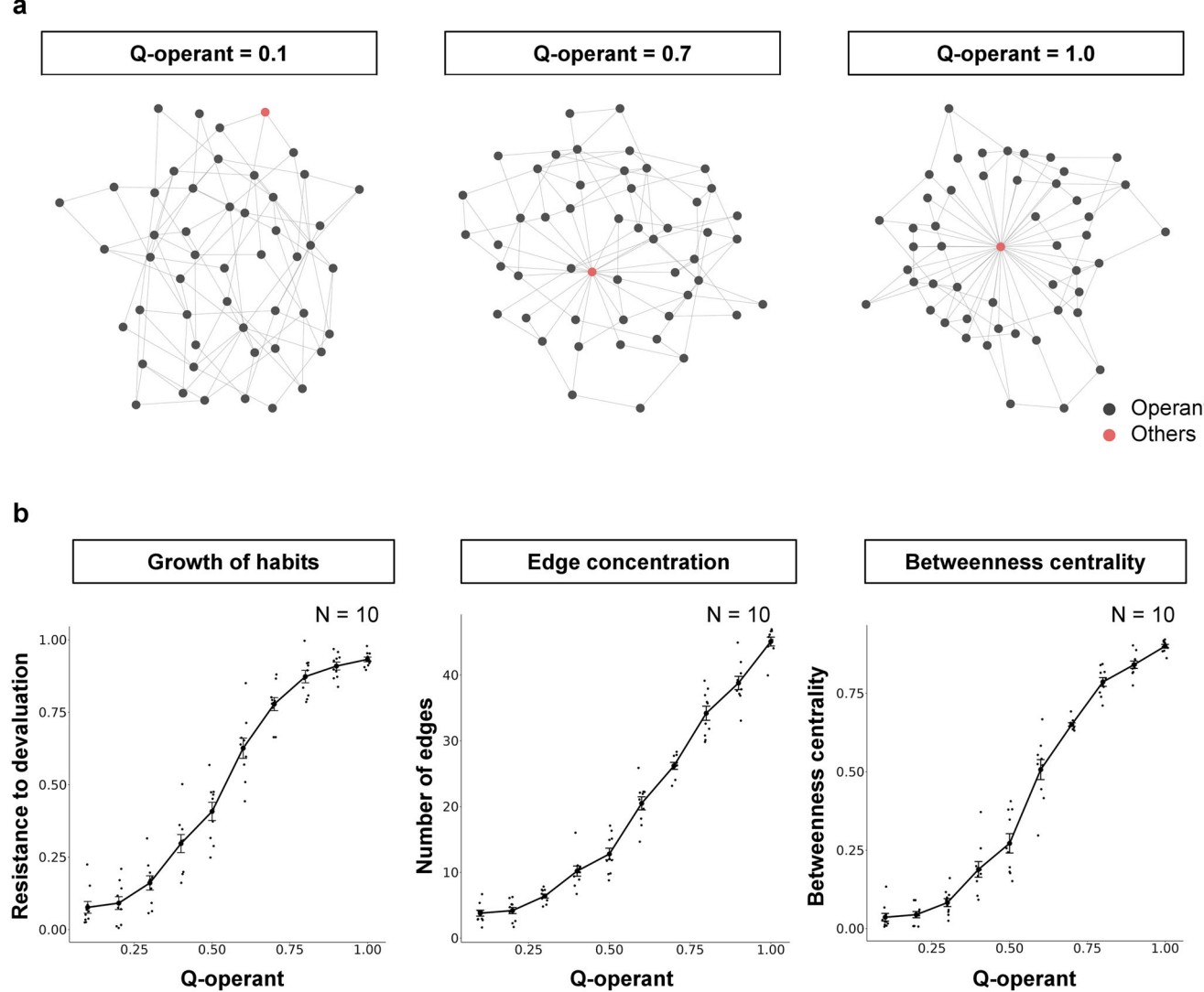

**Fig. 2 Results of simulation 1. a** Change in network with an increased Q-operant. Each point denotes a response, with black and red indicating other responses and the operant response, respectively. **b** Change in resistance to devaluation and features of the network with an increased Q-operant. The left panel shows resistance to devaluation, which indicates the decrease in the operant response caused by reward devaluation and implies that the operant response becomes a habit at higher values. The center panel shows the change in the number of edges that the operant response acquired. The right panel shows the betweenness centrality, i.e., the probability that the operant response is included in the shortest path connecting two nodes in the generated network. Each point denotes individual agent and error bars show standard error of the mean (SEM).

response. After the agent reaches the response, it repeats this process again. After several loops, we calculated the proportion of the operant response to the total number of responses to assess whether the operant response is habit or not.

**Simulation result**. Figure 2a shows examples of generated networks under the Q-operant. Other responses (black nodes) connected to the operant response (red node) as the Q-operant increased. Figure 2b shows the resistance to devaluation (left panel), number of edges that the operant response acquired (center panel), and betweenness centrality (right panel). The resistance to devaluation was larger when the operant response did not decrease with reward devaluation and higher Q-operant the resistance to devaluation were larger (Fig. 2b left). The number of edges that the operant response acquired increased as the Q-operant increased (Fig. 2a and b center), implying that edges from other responses were concentrated to the operant response. The betweenness centrality, i.e., the probability that the operant response is included in the shortest path between two

nodes in the network, increased as the Q-operant increased (Fig. 2b right). With edge concentration in the operant response, distances between two nodes in the network decreased (Fig. 3 left). Furthermore, transitions made by agents in the simulations became efficient, and time required for simulations shortened (Fig. 3 right). These results were replicated in a wide range of Q-operant and Q-others, in different numbers of nodes (Supplementary Fig. 2), and with a different path search algorithm (Supplementary Fig. 3).

**Interim discussion**. In simulation 1, we examined the structure of the network and habit formation under arbitrary Q-matrix and showed that habit formation occurred when edges from other responses were concentrated in the operant response. By manipulating $Q_{operant}$ systematically, the operant response acquired most edges in the network (Fig. 2a and b center) and it caused that increase in the resistance to devaluation (Fig. 2b left). These results suggest that habit formation can be viewed as the structural change in the behavioral network. In particular, habits

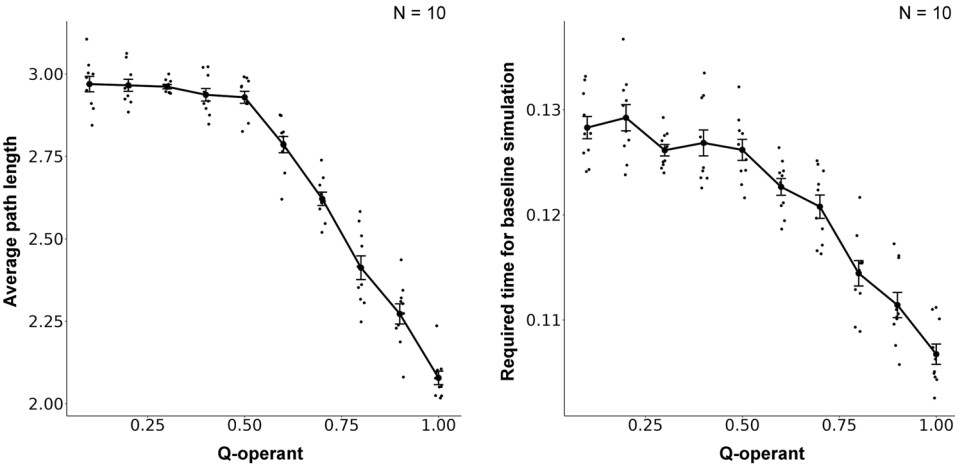

**Fig. 3 Reduced computation costs with habit formation.** The left panel shows the average path length, i.e., the average of the shortest path between two nodes in the network. When the path length is shorter, the transition from one response to another becomes faster. The right panel shows the required time to simulate the baseline phase. The required time is the real time, i.e., the duration from the start to the end of the simulation. Since the number of loops is the same for all simulations, the decrease in required time implies efficiency in shortest path search and transitions between responses. Each point denotes individual agent and error bars show SEM.

are considered as concentration of edges from other responses to the operant response. This is because when agents move one response to another, the operant response is included in the path between the two nodes (Fig. 2b right). These results were replicated in different settings of algorithms or parameters (Supplementary Fig. 1, 2, and 3), suggesting these results were not limited to the specific setting.

Habits are considered to be efficient in the computational cost and transition[7,37]. In our model, these features of habits were also found. Animal responses are constrained by some factors, such as space and the animal's body. For example, an animal cannot eat food if the food is not in front of it and if it cannot walk when it is sleeping. These examples imply that not all responses are connected to each other and that the number of edges in the network is limited. When the number of edges was constrained, the structure of the network promoted that agent to engage in the desired response. When edges from other responses were concentrated in the operant response, the average distance between two nodes was shortened[38], and transitions made by agents became efficient (Fig. 3). These results also imply that agents can find the path between two nodes faster. Thus, habit formation, i.e., edge concentration to the response, reduces the computational cost and hastens the transition under constraints.

**Simulation 2: devaluation and its effect on behavior under free-operant situation.** We examined if our model could reproduce the effects of factors that promote or disrupt habit formation in free-operant situations[11–15]. In simulation 2, we let an agent learn Q-values under arbitrary experimental environments and examine whether habit formation occurs. Under free-operant situations, there are three factors that lead to an operant response to habit. The first is the amount of training, where one response is rewarded repeatedly under one situation, and the response becomes habit[11,12]. The second factor is the rule, called schedule of reinforcement, which determines the criteria for presentation of a reward for a response[13]. Habit formation does not occur when reward presentation is determined by the number of responses by the animal. In this environment, the presence/absence of a reward is determined with a certain probability each time the animal presses a lever, e.g., in the bandit task or slot machine use. Habit formation occurs when rewards are determined according to the time elapsed since the previous reward. In

this environment, the availability of a reward is determined potentially at arbitrary time steps with a certain probability, and the reward is presented at the first response after reward presentation becomes possible, such as checking a mailbox. The former response-based rule is called the variable ratio (VR) schedule, and the latter time-based rule is called the variable interval (VI) schedule. The third factor is the presence of alternatives. If two alternatives are available under a situation and different rewards are obtained from them (e.g., left lever → food, right lever → water), the operant response does not become a habit[14,15]. Here, we reproduce the above experimental settings and examine whether our model becomes a habit under these environments.

The only difference between simulations 1 and 2 is whether the agent learns the Q-values. Here, the agent experienced the training phase preceding the baseline phase, where the agent learned Q-values through interaction with a given environment and constructed a network based on them (more detail in "Materials and Methods"). After the training phase, the agent experienced the baseline, devaluation, and post devaluation phases in the same way as in Simulation 1.

**Simulation result**. Figure 4a shows the growth of resistance to devaluation (left), number of edges (center), and betweenness centrality (right) with increased amounts of training in VI (time-based rule; red line) and VR (response-based rule; blue line) schedules. All measures were larger in the VI schedule than in the VR schedule. Figure 4b shows the resistance to devaluation (left) and examples of networks learned in the choice (center) and no-choice situations (right). The resistance to devaluation was larger in the no-choice situation than in the choice situation (Fig. 4b left). Two operant responses acquired almost the same number of edges in the choice situation (Fig. 4b center), while only one operant response acquired the most number of edges in the network in the no-choice situation (Fig. 4b right). Figure 5 shows the Q-value for self-transition of the operant response. The Q-value increased with an increased amount of training and was larger in the VR schedule than in the VI schedule. These results were replicated in different experimental settings. Supplementary Fig. 2B shows the replicated results in different numbers of nodes (25, 50, 75, and 100). In simulation 2, agents received rewards every time they engaged in other responses. In other words, we

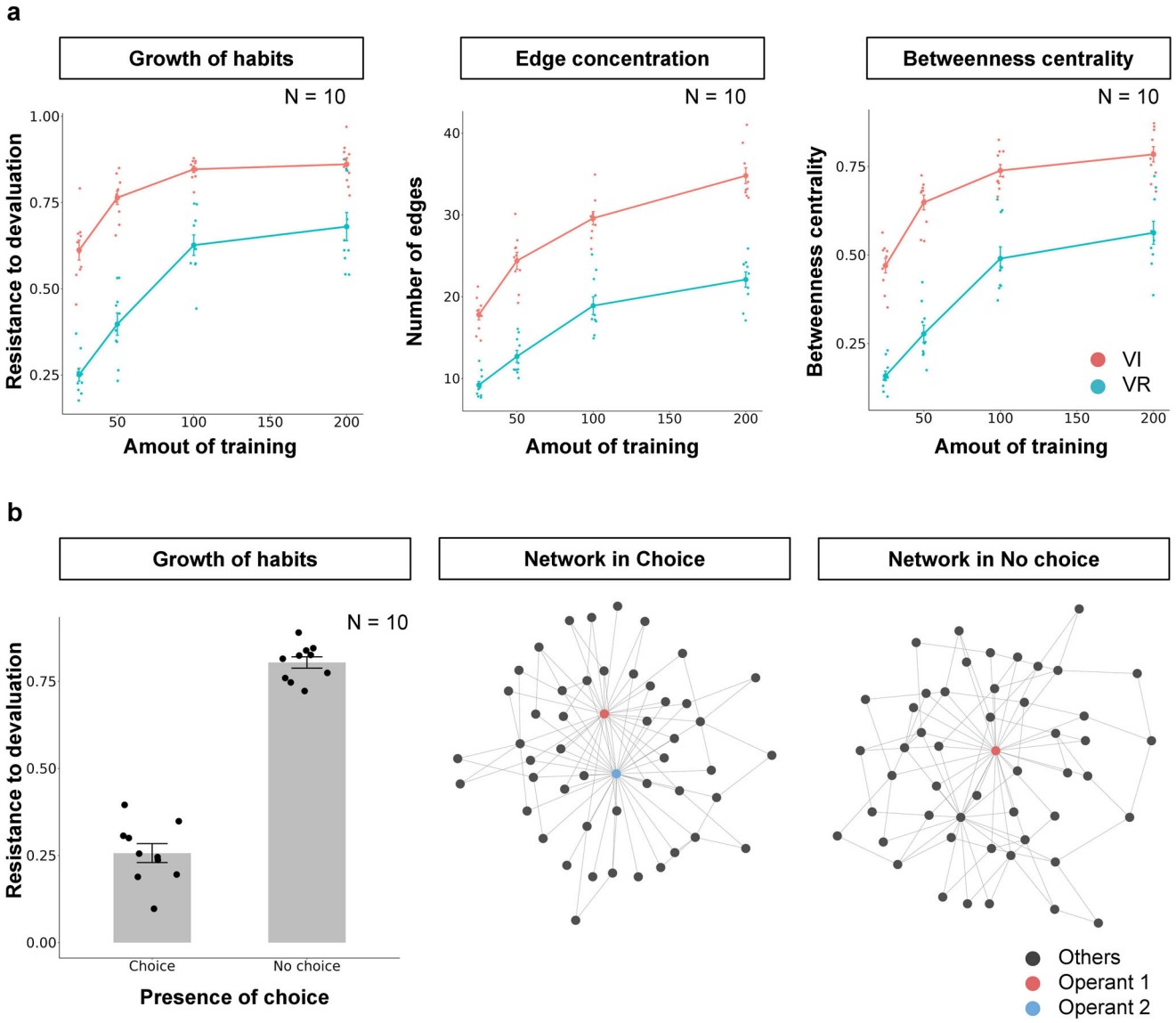

**Fig. 4 Results of simulations in VI and VR schedules and presence and absence of choice. a** Growth of habits and network features in simulations manipulating the amount of training in the VI and VR schedules. In all panels, the red and blue lines denote the VI and VR schedules, respectively. The left, center, and right panels show the resistance to devaluation, number of edges, and betweenness centrality, respectively. **b** Effects of choice on habit formation and network features. The left panel shows the resistance to devaluation. The center and right panels show the learned network in the choice and no-choice situations, respectively. In the network, the red and blue nodes denote the operant response, and black nodes denote other responses. Each point denotes individual agent and error bars show SEM.

assigned fixed ratio (FR) 1 for other responses. Supplementary Fig. 4 shows the results when a different schedule was assigned to other responses instead of FR 1. The results were almost the same. We examined if the results remained similar when a different learning algorithm, SARSA, was employed and Supplementary Fig. 5 shows that similar results were obtained.

**Interim discussion.** In simulation 2, we examined whether our model shows similar behavior to real animals in environments that affect habit formation, and our model reproduced the similar results reported from the empirical studies. The resistance to devaluation increased with an increased amount of training and was larger in the VI schedule than in the VR schedule (Fig. 4a left). As we have seen in simulation 1, the operant response acquired most of the edges in the network under VI schedule, but not under VR schedule (Fig. 4a center), and it turned out that the betweenness centrality grew up under VI schedule (Fig. 4a right).

These results imply that the VI schedule and a large amount of training promote habit formation. The resistance to devaluation was lower in the choice situation than in the no-choice situation (Fig. 4b left), suggesting that the presence of explicit alternatives disturbed habit formation.

The amount of training affects the structure of the network (Fig. 4a), and as the amount of training increases, the cohesion of edges in the operant response increases. The smaller the amount of training, the smaller the Q-values of the transition from other responses to the operant response. Consequently, the probability that an edge is attached to the operant response is smaller. As shown in simulation 1, habit formation occurs when the operant response acquires most of the edges in the network. Thus, the amount of training affects habit formation.

The resistance to devaluation was larger in the VI schedule than in the VR schedule, suggesting that habit formation was promoted in the VI schedule. The VR schedule is a response-

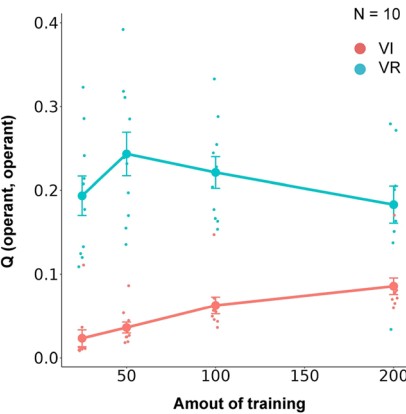

**Fig. 5 Q-value for self-transition of the operant response.** Q-value of self-transition of the operant response. The red and blue lines denote the VI and VR schedules, respectively. Each point denotes individual agent and error bars show SEM.

based rule of reward presentation. Therefore, all operant responses, independent of the agent's engagement immediately before, were rewarded with constant probability. In contrast, the VI schedule is a time-based rule and it causes that an operant response, longer elapsed time from last operant response, is selectively rewarded. In other words, an operant response emitted after a few periods was selectively rewarded and implied a transition from the other responses to the operant response in our model. In summary, transitioning from other responses to the operant response was selectively rewarded in the VI schedule and resulted in edge concentration in the operant response and habit formation.

One might suspect that, contrary to the experimental facts that the response rate is larger in VR schedule than VI schedule, if operant responses acquire more edge in the VI schedule, then response rate would be higher in the VI schedule as well. However, Fig. 5 shows the Q value of the self transition of the operant response is larger in VR schedule than VI schedule. It implies that once an agent starts to engage in an operant response, it will repeat the same response over and over again. In fact, it has been experimentally shown that the difference in response rate between VI and VR schedules is caused by such a mechanism[39–41].

Although the operant response acquired most of the edges on the network under the choice environment, the operant response did not become a habit. There are two reasons for this. First, the agent chooses its response based on the value of the reward obtained from the response. In the test phase, the value of the reward obtained from the operant response was reduced, and that of the alternative response remained the same value as the baseline. Thus, the agent chose the alternative response more in the test phase than in the baseline phase. Second, if only the operant response acquired most edges, any shortest path may contain the operant response. However, the alternative response acquired most of the edges, so that any shortest path contained the alternative response. Thus, the operant response no longer has a greater chance of being engaged, and habit formation does not occur.

In the no-choice situation, the operant response acquired the most edges in the network, but several other responses also acquired multiple edges (Fig. 4b right), resembling the scale-free network, which should be assessed by the distribution of degree. However, habit formation occurred in the network. Therefore, although scale-free networks were not compared with random or hub-and-spoke networks, habit formation might be present in the scale-free-like network.

**Simulation 3: correlation-based account vs contiguity-based account of habit formation.** Here, we propose an experiment to directly test the response-reinforcer correlation, which has been considered as a factor leading to habit formation in the past, and our model's explanation: selective reinforcement of transitions from other behaviors to the operant response and the resulting structural changes in the network. This is a new experiment predicted by our model, which has not yet been examined in real animals, and will encourage future theoretical tests.

From canonical view, response-reward correlation, the operant responses remain goal-directed when animals experience a correlation between the operant responses and rewards, but become habits when they do not experience the correlation[1,42]. Under VR schedule, the more they engage the operant response, the more rewards they can obtain. It leads that they experience positive correlation between the operant response and rewards, and the operant response remains goal-directed. In contrast, under VI schedule, since rewards availability is governed by time, such correlation is collapsed and they do not experience it. It results that the operant response becomes habit.

In recent years, results have been reported that contradict the response-reward correlation[43–45]. For example, De Russo, et al.[44] trained mice under VI and FI schedules. FI and VI have a common molar relationship between response rate and rewards: in both schedules, animals can not obtain more than the determined number of rewards within a certain duration, no matter how much they engage in the operant response. Under such a condition, the response-reward correlation view predicts that both schedules guide the same level of habit formation. However, the operant response of mice trained under FI schedule remains goal-directed but under VI schedules, the operant response becomes habit. DeRusso, et al.[44] conclude that the contiguity, which is defined by average temporal distance between responses and successive rewards, disrupts habit formation. In the FI schedule, animals tend to emit more response as they approach the time when rewards are presented. In contrast, animals do not know when the reward becomes available, they emit responses uniformly during inter-reward intervals in VI schedule. Thus, under the FI schedule, animals emit many responses just before rewards and the contiguity of responses and rewards becomes higher but, under the VI schedule, operant responses are distributed uniformly and the contiguity becomes lower.

A similar discussion has been made for VI-VR response rate difference and there are two kinds of accounts. One explains the difference by the difference in interresponse time that is likely to be rewarded[46,47]. In VI schedule, probability of reward availability increases as the elapsed time from last response increases and it results that longer IRTs are more likely to be rewarded than shorter ones. In contrast, such characteristics are not found in the VR schedule or shorter IRTs are more likely to be reinforced. (Fig. 6 right). Thus, response rate is lower in VI schedule than VR schedule. Especially, the copyist model[46] explains the difference by average of interresponse times between successive rewards and this is similar to contiguity-based account of habit formation[44,45]. Second account is based on the molar relationship between response rate and reward rate[48,49]. The more animals emit responses under VR schedule, the more rewards they can obtain (blue line in Fig. 6 left). In contrast, under VI schedule, animals can not obtain more rewards than experimentally defined, no matter how they emit responses under the schedule (red line in Fig. 6 left). This account underlies the response-reward correlation account of habit formation[1,42].

Our model is positioned similarly to the contiguity-based account in these discussions. As we show in simulation 2, the VI-VR response rate difference can be explained by which transitions are likely to be rewarded: In VI schedule, the transitions from

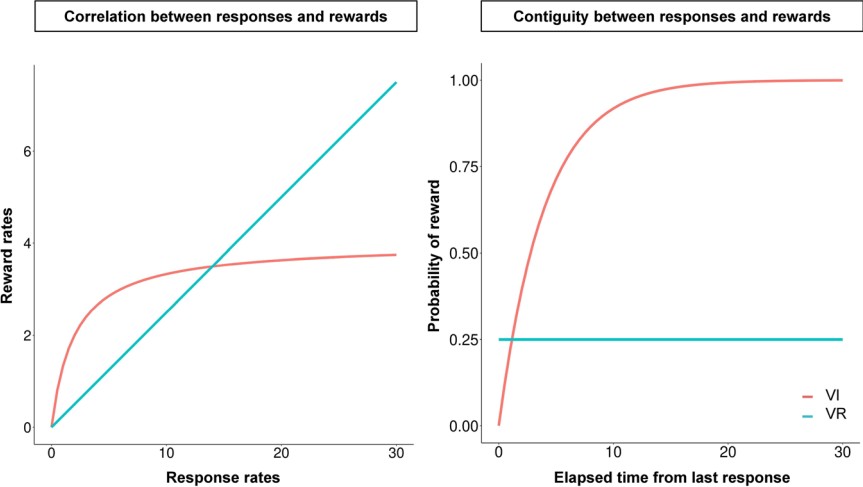

**Fig. 6 Example of response-reward correlation and contiguity.** Response rate and reward rate correlation (left) and reward probability as function of elapsed time last response (right) in VR and VI schedules. In VR schedule (black line), reward rate is proportional to response rate, in contrast, reward rates reach a plateau as response rate increases in VI schedule (red line). Reward probability is constant independent from elapsed time from last response in VR schedule, in contrast, it increases exponentially as the time increases.

other responses to the operant response are more likely to be rewarded but not in VR schedule (Fig. 5). Viewing the cause of long IRTs as engagement in other responses[32,50], differential reinforcement of long IRTs can be interpreted as differential reinforcement of the transition from other response to the operant response. Considering these discussions, our model suggests that the same discussions for VI-VR response rate difference can be applied to habit formation.

Here, we mimic an experiment which is conducted to reveal that the VI-VR response rate difference is caused by IRTs immediately followed by rewards[51]. In the experiment, pigeons are trained under tandem VI VR and tandem VR VI schedules. The former schedule, tandem VI VR, shares a molar relationship between response rate and reward rate with VI schedule. However, VI schedule is immediately followed by short VR schedule and longer IRTs are less likely to be rewarded than simple VI schedule. The later one is tandem VR VI, it's molar relationship between response rate and reward rate is similar to the simple VR schedule. However, since VR schedule is followed by VI schedule, longer IRTs are more likely to be rewarded. In this schedule, pigeons showed higher response rate in tandem VI VR schedule and lower in tandem VR VI schedule[51]. These findings contradict the account based on response rate and reward rate correlation but well explained by differential reinforcement of IRTs[46]. Will habit formation occur under these schedules? From the view of response-reward correlation, tandem VI VR schedule leads habit but not in tandem VR VI schedule because there is lower response-reward correlation under the former schedule but higher than the later one. In contrast, our model makes the opposite prediction that habit formation will be guided under tandem VR VI schedule but not under tandem VI VR schedule. This is because, in the former schedule, transitions from other responses to the operant response are more likely to be rewarded, and the operant response acquired more edges. In the later schedule, transitions from other response to the operant response and the self transition of the operant response are rewarded in the same probability so the operant response acquired not so many edges.

**Simulation result**. Figure 7 shows the resistance to devaluation, number of edges, and betweenness centrality simulated under VI, tandem VI VR, VR, and tandem VR VI schedules. They were

higher under VI and tandem VR VI schedules than VR and tandem VI VR schedules. Although the response-reward correlation account suggests that habit formation is disrupted under tandem VR VI schedule and is promoted tandem VI VR schedule, the results were the opposite, habit formation was promoted under tandem VR VI and but not under tandem VI VR. The center of Fig. 7 shows the number of edges that the operant response acquired to the overall number of edges in the network and the operant response acquired more edges under VI and tandem VR VI schedules. Figure 7 (right panel) shows the betweenness centrality of the operant response. The betweenness centrality was larger in the VI and tandem VR VI schedules than in the VR and tandem VI VR schedules. Figure 8 shows the Q-value of the operant response. It was larger in the VR and tandem VI VR schedules than in the VI and tandem VR VI schedules.

**Interim discussion**. In simulation 3, we mimicked the schedules employed by Peele et al.[51] to reveal what characteristics of schedules, response-reward correlation or response reward contiguity, promote habit formation. Traditional accounts suggest that lack of the response-reward correlation promotes habit formation[1,42]. In contrast, other researches suggest that the response-reward contiguity is crucial for habit formation but not the correlation[44,45]. These two accounts make different predictions in the schedules we employed here. Tandem VR VI schedule has a common molar relationship between response rate and reward rate with simple VR schedule (blue line in Fig. 6 left) but it also has a time-dependent property, which is found in VI schedule (red line in Fig. 6 right), that the probability of obtaining rewards increases as time elapses. In summary, the Tandem VR VI schedule has higher response-reward correlation but lower response-reward contiguity, and the response-reward correlation account predicts that habit formation is disrupted in the schedule. In contrast, the tandem VI VR schedule lacks both a molar relationship between response rate and reward rates and time-dependency (red line in Fig. 6 left and blue line in Fig. 6 right). In such schedule, animals can not obtain more than the determined number of rewards within a certain duration, no matter how much they engage in the operant response but the transition from other responses to the operant response is less likely rewarded. In summary, the tandem VR VI schedule had a higher

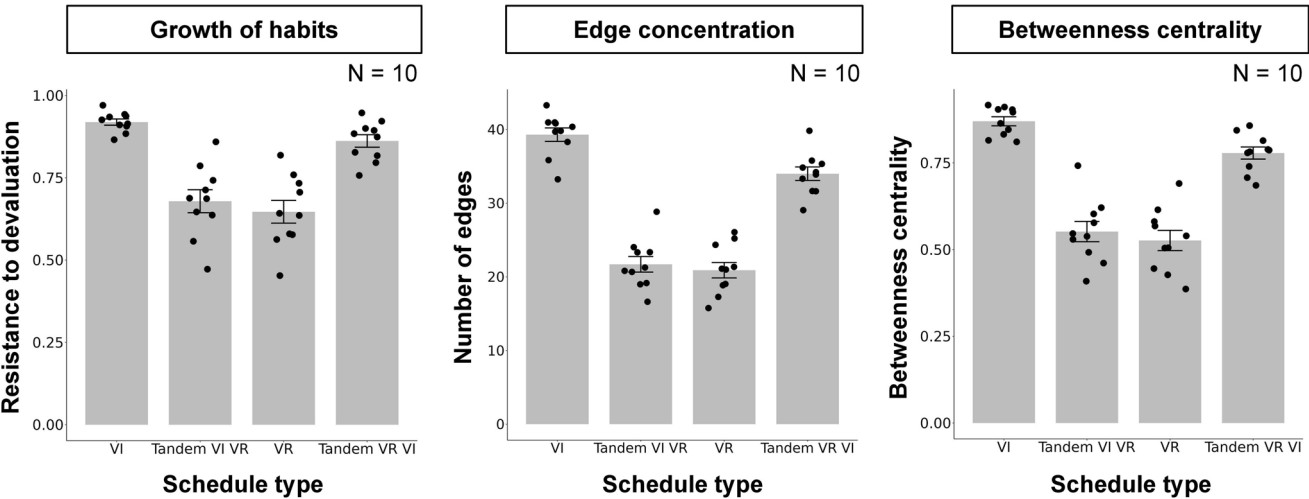

**Fig. 7 The simulation results in tandem VI VR and tandem VR VI schedules.** The left panel shows the resistance to devaluation in VI, Tandem VI VR, VR, and Tandem VR VI schedules. The center panel shows the number of edges that the operant response acquired in each schedule. The right panel shows the centrality of the operant response. Each point denotes individual agent and error bars show SEM.

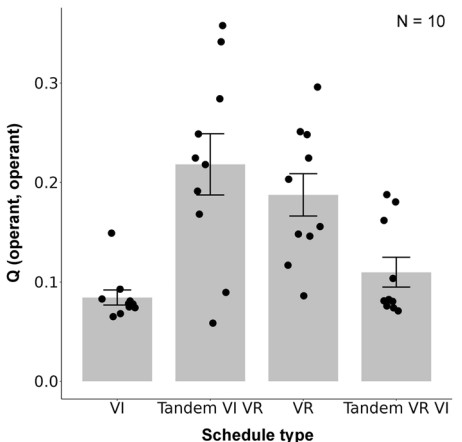

**Fig. 8 Q-value for self-transition of the operant response.** Each point denotes individual agent and error bars show SEM.

response–reward correlation but a lower response-reward contiguity, and the response-reward correlation account predicted that habit formation was disrupted in the schedule. However, in contrast to the traditional view, our model predicts that habit formation is more likely promoted in tandem VR VI schedule (Fig. 7 left). Because of time-dependency tandem VR VI schedule have, transition from others response to the operant response is more likely to be rewarded in the schedule and acquired more edges that simple VR schedule and tandem VI VR schedule (Fig. 7 center). Thus, as we showed in simulation 1 and 2, the probability that the operant response is included in the shortest paths increased and habit formation occurred (Fig. 7 right).

Our model supports the account that the contiguity between responses and rewards promotes habit formation[44,45]. In tandem VI VR and simple VR schedule, the self transition of the operant response is more likely rewarded than transition from other responses to the operant response. This is because, the operant response occurred as a bout, a burst of responses is followed by long pauses, and this implies that animals emit more responses just before reward presentation. In tandem VR VI and simple VI schedule, the self-transition of the operant response is less likely rewarded because of the time-dependent property between response and reward (red line in Fig. 6 right). This result implies

that animals emit less response just before the reward presentation. Thus, response reward contiguity is higher in the tandem VI VR and simple VR schedule than tandem VR VI and simple VI schedule.

## Discussion

In this research, we explain habit formation as changes in network structure by assuming the behavior of organisms viewed as a network of responses. In simulation 1, we generated arbitrary networks and examined the underlying structure of goal-directed behavior and habits. We revealed that habit formation occurs when a particular response acquires most of the edges from other responses. In Simulation 2, we simulated the environments that were reported to promote or inhibit habit formation from existing studies and examined whether the proposed model showed habit formation. These results were consistent with experimental results reported by many laboratories, suggesting that our results demonstrate habit formation as a structural change in the behavioral network. In simulation 3, we analyzed the behavior of the proposed model in an experimental situation where the canonical theory[1,42] and the proposed model make different predictions. The results suggest that our model supports the view of reward-responses contiguity promoting habit formation[44,45] but not the canonical view of reward-response correlation.

**Relationship to other theoretical models of habit formation.** Although there are many models of habit formation, most of them are viewed as goal-directed behavior and habits as interactions between two distinctive associative structures. Here, we succeeded in providing a novel explanation by taking a more molar view of behavior. Specifically, the proposed model substantially differs from existing models in three ways. First, the proposed model does not consider behavior as a single element, but as a network of interconnected responses. Conventional views focus only on responses under highly constrained experimental situations, such as lever pressings or button pushings, and ignore the molar structure of behavior that the real organisms may have. Responses of organisms, including humans, are not independent of each other, but they are probabilistically conditioned by the preceding and succeeding responses. In the proposed model, the structures of such responses are represented as a network, and habit formation is explained as a change in the structure. Second, our model seems to have no state variable, unlike previous

models[3,4,7,52]. We treated the immediately prior response of the agent as a state; thus, so there is no lack of state variables. This treatment of past responses as states has often been employed in modeling animal behavior[31,32,50,53]. However, our model differs from past models of habits. Many models of habits were built in consideration of the multistage Markov decision task[2–4,7]. In the multistage Markov decision task, experimentally explicit states, each choice point, exist. In contrast, we studied habits in free-operant situations in which animals could engage in responses freely and repeatedly, and experimentally explicit states were lacking. Previous models were applied to the free-operant situation in two different ways. One way was to not assume the state[1], and the other way was to introduce a hypothetical state[7,52]. We treated the immediately prior response as a state, similar to the later one. Although our model seems to have no state variable, our approach was similar to the previous one[7,52]. Thus, we suggest that the inclusion or exclusion of state variables to explain habit formation in free-operant situations depends on the details of the model and is not always necessary. Third, some models of habits assumed two distinct systems corresponding to goal-directed behavior and habits[1,2,52]. Particularly, only the model that could explain habits in free-operant situations assumed them explicitly[1]. Although all responses were assumed to be under goal-directed control, choices were based on reward values and shortest path search, and results reported in free-operant situations were reproduced[11–15]. Recently, in the context of the multistage Markov decision task, several models showed no distinct systems between goal-directed behavior and habits[3,4]. Our model also showed no explicit distinction but that the idea could be applied to habits in free-operant situations.

Although the proposed model deals with experiments on habit formation in rodents' operant situations[11–15], most of the experiments discussed here are also dealt in Perez and Dickinson[1]. Both models reproduce results that are consistent with the experimental results. Perez and Dickison[1] provide an explanation based on reward-response correlations. In their model, the lower the correlation between response and reward, the more habit formation is promoted. On the other hand, the proposed model provides an explanation based on contiguity between response and reward[44,45]. Contiguity is defined by temporal distance between the reward and the emitted response to obtain it. The lower the contiguity, the longer the temporal distance between the response and the reward, the more habit formation is promoted. Although the proposed model does not explicitly incorporate contiguity as a variable in the model, it allows for a similar representation by dividing the agent's behavior into the operant responses and other responses, and separating transitions to the operant responses into self-transitions and transitions from other responses. For example, in a schedule with low reward-response contiguity, such as the VI schedule, transitions from other behaviors to the operant are more likely to be reinforced, while in a VR schedule with high contiguity, transitions from other behaviors are less likely to be reinforced. As a result, the operant response obtains more edges and promotes habit formation in schedules with low contiguity. As an experiment in which these two factors can be more clearly separated, we employed the procedure of Peele et al.[51]. Under this procedure, correlation-based and contiguity-based explanations provide opposite predictions. The proposed model reproduced the same results as predicted by the contiguity-based explanation. Whether habit formation occurs under this experimental procedure has yet to be examined, but it does provide useful insights for updating the theory of habit formation.

The proposed model may seem similar to the model of Dezfouli and Balleine[7,9,10]. In fact, their model and our proposed model have two common assumptions. First, instead of treating the agent's behavior as a single response, the two models explicitly assume other responses. They explain habit formation in terms of the acquisition of those sequences or the structure of the network. The second point is that the agent generates sequences or searches for the shortest path based on the value of the reward. However, the models have two differences. First, the targeting experimental situations differed. Their model was built with the multistage Markov decision task, while our model was built to explain habit formation in free-operant situations. The existing comprehensive theory in free-operant situations assumed parallel control by two systems (1). A kind of response-chaining/action-chunking models have limited applicability in free-operant situations. Second, the view of behavior differed. Our model tried to overcome the limitation. In free-operant situations, animals could engage in responses freely without explicit states defined experimentally. In the case of free-operant situations, direct application of the idea of response-chaining or action-chunking was difficult because no points corresponded to the start and end of trials. Instead of the chunk or chain, we considered behavior as a network and the agent's behavior as a transition within the network. In other words, by viewing behavior as a loop without a clear start or end, we successfully modeled the behavior of free-operant situations.

Actually, Dezfouli and Balline[7] applied their model to the free operant situation and reproduced the effect of amount of training on habit formation. However, they did not treat how other factors, schedule types and presence of alternatives, affect habit formation. The proposed model, which shares common assumptions with their model, can reproduce the results reported in empirical study[11–15], suggesting that the idea of response-chaining or action-chunking could be applied in free-operant situations. Moreover, the model clarifies the difference between the canonical correlation-based account and common points with the contiguity-based account. We also found common features with the recently proposed models[3,4]. In those models, goal-directed planning was employed, and the behavior of human and rodents' multistage decision-making tasks, such as multistage Markov decision tasks and tree-shaped maze, were explained. Pezzulo et al.[3] built a mixed-controller model consisting of goal-directed and habit behaviors in a single system. Keramati et al.[4] proposed that these two systems were not separated but placed in one spectrum. Our model also considered these two systems to be not separated but coexisting in a single system and placed in one spectrum, with only a difference in the structure of the network. However, similar to many other models, their models targeted multistage decision-making tasks but not free-operant situations. Our model shared common features, i.e., planning and singularity of the system, with their models[3,4] and successfully applied those features in free-operant situations. From the canonical view, two distinct systems control a response in the flat manner[1,2]. This view has been challenged recently, and new models have been proposed in the context of the multistage decision-making tasks. Although their applications are limited to free-operant situations, our model adopted those ideas, i.e., response-chaining/action-chunking, planning, and mono-systematicity, and explained habit formation in free-operant situations, suggesting a link between the different experimental procedures and providing a comprehensive understanding of habit formation.

**Neural substrates of behavioral network**. The corticostriatal network is involved in habit formation, and generates response patterns[54,55]. Especially, dorsolateral striatum (DLS) is known to be important in transition from goal-directed behavior to habits[56]. DLS activity changes as proceedings of training and responses become habits[57,58], and lesion of DLS turns habits into

goal-directed behavior after extended training[56]. DLS also carries forming response sequences[59] and motor routines[60]. In addition to its importance in the learned behavior, DLS also encodes innate response sequences[61]. These facts imply that habit formation and the formation of response sequences have common neural substrates.

A recent study reported that DLS encodes not only information about response sequences but also more divergent information about behavior, which are topographically categorized responses and transitions between them[17]. They recorded the DLS activities of mice with fiber-photometry under an open-field situation and reported neural activities that correlated with the behavior. The activities differed depending on the preceding and succeeding responses, and DLS encoded a transition between the responses. Moreover, the behavior of the mice with DLS lesions showed random transitions of the responses compared to the sham-lesion group. These results imply that the information encoded in DLS is the transition of the structure of behavior. Thus, the function of the DLS might be well understood by considering the habit and goal-directed behavior from the viewpoint of the behavioral network.

Corticostriatal circuits, the associative network, which consists of the prefrontal cortex, dorsomedial, or ventral striatum, plays a role in goal-directed behavior[62]. The dorsomedial striatum (DMS) is known to be involved in the acquisition of goal-directed behavior, maintaining sensitivity to outcomes, and expressing goal-directed behavior[63,64]. The DMS receives excitatory inputs from the prefrontal cortex, whereas the DLS receives inputs from the sensorimotor and premotor cortices[64]. In the canonical dichotomous view of habit formation, goal-directed behavior is replaced by habit after extensive training. After habit formation, the contribution of DLS becomes more important than that of DMS[56,64]. However, even after extensive training, many brain areas such as the prefrontal cortex, anterior cingulate cortex, and ventral and dorsal striatum are modulated by anticipated rewards[65–69]. In our model, any response emitted by an agent is considered goal-directed. Regardless of the training stage, our agents choose their responses based on the value of the rewards. Therefore, the fact that regions involving goal-directed behavior are modulated by anticipated rewards even after extensive training, our assumptions do not contradict each other. Combined with the fact that DLS is more responsible for sequential responding than DMS[70], the transition from DMS to DLS during habit formation might reflect the corresponding behavioral sequence induced by changes in the behavioral network.

Neuronal circuits involving ventral striatum and hippocampus play key roles in spatial navigation and are considered to be related to the planning[71,72]. Both spatial navigation and planning are related to habits and they share common neurobiological substrates[3,73–77]. Although roles of hippocampus and planning in habits and goal-directed behavior in free-operant situations remains unknown, our model sheds light on the role of planning and related brain regions in habits in the free-operant situations.

**Relationship to other behavioral phenomena**. Animals engage in specific responses, such as orienting, approaching, and consummatory behavior, just after the presentation of the reward. Specific action sequences are observed during experiments, and learning is sometimes disrupted by innate responses. These experimental and observational facts lead us to assume that behavior is a network constructed from responses.

In our model, the structure of a network depends only on past experiences under a given situation. In other words, our model does not consider the connections between specific responses that

real organisms may have. Thus, we could not reproduce this phenomenon. However, our model can be further extended and modified to include this phenomenon.

Schedule-induced behavior, observed under intermittent schedules of reinforcement, is a behavioral phenomenon in which animals show aggression or water intake just after the reward presentation[25–27]. This phenomenon can be attributed to the innate connections between reward consumption and schedule-induced behavior. Because of these connections, animals tend to engage in aggression or water intake immediately after reward presentation. Similarly, terminal behavior, which occurs as approaches reward presentations, can be explained by assuming an innate connection, which may explain the fact that animals show a specific sequence of responses during the experiment.

To deal with such phenomena, we assume that it is possible to express the innate susceptibility of edges as a prior distribution and impose constraints on the probabilities of edges attached by learning. Furthermore, we can systematically treat phenomena such as misbehavior and biological constraints on learning by examining differences in prior distributions among species and environments. Thus, we can extend our model to a comprehensive framework of behavior that incorporates the innate behavior of organisms under natural settings.

Goal-directed behavior and habits are related to spatial navigation[3,73–77]. Pezzulo et al. [3] target an experiment with tree type maze and the task is similar in the abstract structures to the multistage Markov decision task. Our model employed a planning process as the model proposed in Pezzulo et al. [3]. However, planning is made in the real space in their model, but planning is made in behavioral space in ours. Thus, the application of our model for spatial navigation is limited. However, the idea of learning response sequence can be applied to spatial navigation, such as learning a series of responses of turning to left and then turning to the right. As we discussed in the above, the limitation is also related to the experimental situations, multistage Markov decision tasks and free-operant situations. We expect a more comprehensive view or model that targets both experimental situations in the future.

**Limitations and future directions**. Our model has three major limitations. First, as we discussed in the previous section, our model does not consider innate constraints that real organisms have, and we believe that we can solve the problem by expressing the innate constraints as a prior distribution. Second, our model could not treat the self-transition of each response. Third, it can only deal with experiments on habit formation under free-operant situations.

Our model cannot treat the self-transition of responses because we employed the shortest path search algorithm to generate response sequences. Any self-transition makes paths between any two responses longer, and paths containing self-transitions must not be the shortest paths. However, animals show a particular response pattern, which is called bout-and-pause and characterized by phasic bursts of one response and pauses following them. Such response patterns imply that the responses have self-transitions. To solve this problem, it is necessary to employ a different algorithm to generate response sequences that allow self-transition.

All our simulations deal with experiments in free-operant situations, and not with recent experiments with the two-stage Markov decision task. This is not a specific problem for our model; other existing models treat either of them. Although many experiments have been conducted in both experimental tasks, the differences and identities of the procedures and results among them have not been systematically examined. To obtain a more

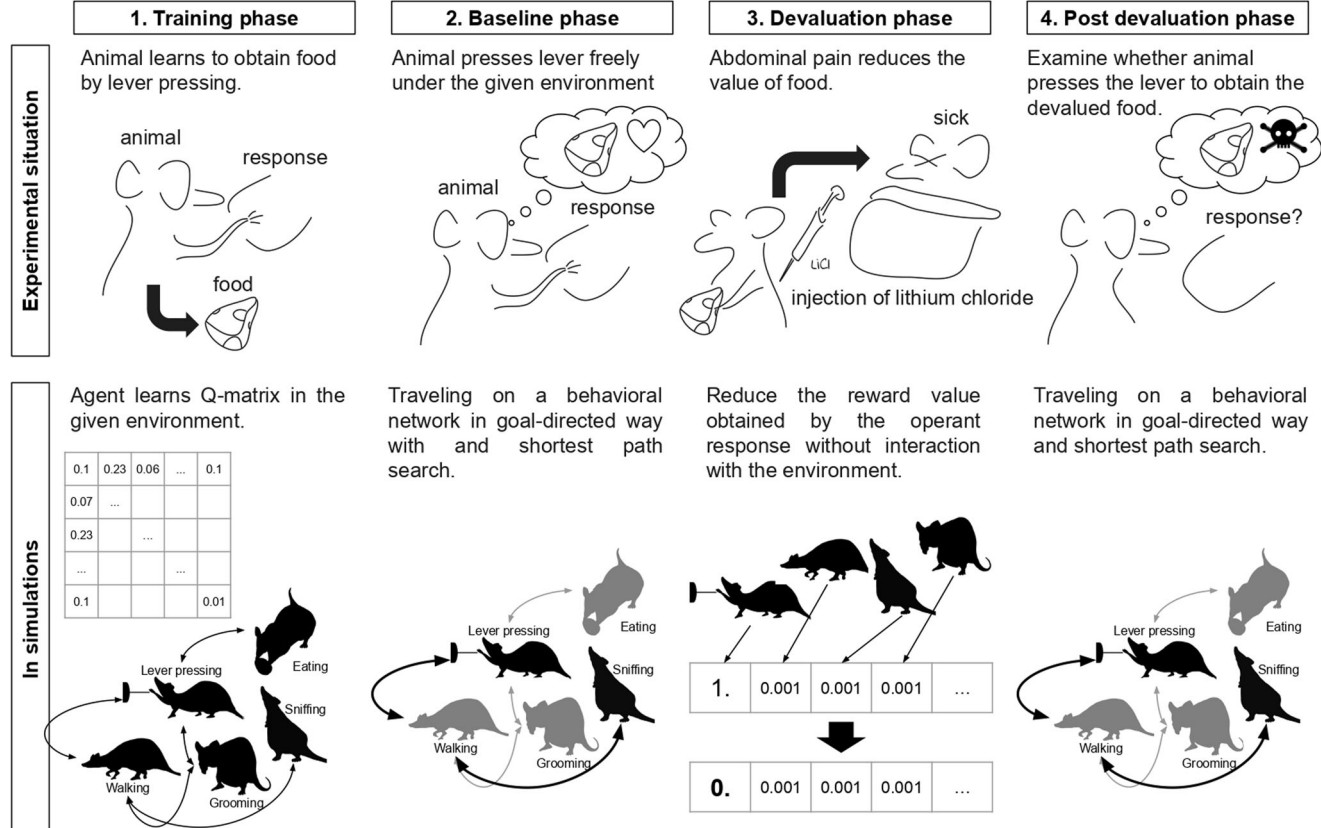

**Fig. 9 Overview of simulations.** The upper panel shows the schematic representation of the reward devaluation procedure in real experiments. The bottom panel shows the simulation procedure as corresponding to the empirical procedure.

unified understanding of habit formation, we need to conduct a systematic analysis of the procedures and results employed and obtained from existing studies. Therefore, the validation of our model is limited to habit formation in free-operant situations.

Recent advances in machine learning allow us to measure animal behavior more objectively and precisely than ever before. However, behavior estimation technologies are not well established at present, preventing us from validating some assumptions in our model. In this field, no consensus has been reached on what timescale should be employed to classify behavior and how finely behavior should be classified. For example, we assumed that the behavioral network consisted of 50 nodes but did not know how many nodes constitute the behavioral network of real animals. However, as shown in Supplementary Fig. 2, habit formation occurred in networks of a slightly smaller size, suggesting that our explanation for habits could be applicable to the real behavioral network even if the size is smaller than we assumed. In the future, such technologies and by utilizing these techniques, it is possible to understand behavior on a macroscale rather than capture the behavior in highly constrained experimental settings. Our model provided a novel perspective on how behavior could be viewed on macroscale behavioral phenomena and raised questions that could be answered by such techniques, which would further help us understand the function of the brain in behavioral changes.

In this paper, we provide a novel perspective on habit formation by assuming behavior as a network. In existing models, goal-directed behavior and habits are controlled by two distinct systems. On the other hand, our model shows that although all responses are goal-directed, both goal-directed and habits result from the structure of the network. It proposes that habit formation is not caused by a change in the control of the two systems, but rather by a continuous change in a single system. Furthermore, the most important feature of our model, which differentiates it from other models, is that behavior is a network constructed from responses. With this view, we have succeeded in providing a novel explanation for habit formation. This implies that the possible algorithms can be changed depending on how one views the behavior of organisms. Our study also suggests that changing the method of capturing behavior could be a fundamental step in understanding the biological structure of the behavior.

## Methods

**Overview**. We conducted three simulations in this article, and they contain four steps (Fig. 9). In the first step, agents learn the Q matrix in the given environments in the simulation 2 and 3 but agents are given a hypothetical Q matrix in the simulation 1. After the training phase, the agents generate a network based on the Q matrix. The way to generate the network is the same in all simulations. In the second step, the baseline phase, the agents travel in the network and engage responses. Here, the agents choose their responses based on reward values and the reward value obtained by the operant response is set to 1.0. The agents no longer update the Q matrix nor reconstruct the network. In the third step, the devaluation phase, the reward value of the operant response ($r_0$ in Eq. 4) was reduced from 1.0 to 0.0 without any interaction with environments. If there were two operant responses, the value of only one of them (reduced $r_0$ but not $r_1$) was reduced. In the fourth step, the post devaluation phase, the agents behave in the same way as the baseline. However, the reward value of operant response is reduced to 0.0. The only difference between baseline and devaluation is the reward value of operant response. We explain the procedures conducted in the four steps in detail after sections. Our simulation codes are available at: https://github.com/7cm-diameter/hbtnet.

**Generate hypothetical Q matrix**. Here, the agents are given a hypothetical Q-matrix instead of learning it through interactions between an environment. First,

we determine the number of nodes contained in a network. We assign a scalar value for each node and it is represented by a vector. The first element of the vector denotes the operant response and other elements denote other responses. The values of the other responses are fixed to 0.01. The value of the operant response is ranged from 0.01 to 1.0. The Q matrix is then defined as the direct product of the Q-vector.

**Learn Q matrix in the given environment**. In simulation 2 and 3, agents learn the Q-matrix in an experimental environment. In the simulation 2, we conducted simulations with variable interval (VI), variable ratio (VR), concurrent VI VI, and VI with non-contingent rewards. Moreover, we changed the number of rewards in the learning phase to examine the effect of training on habit formation. In simulation 3, we conducted simulations with tandem VI VR and tandem VR VI.

In all these simulations, the agent chooses a response and the environment provides a reward based on the response. The agent chooses a response according to the softmax function; $p_i = \frac{e^{\beta_c Q_i}}{\sum_{j=1}^{N} e^{\beta_c Q_i}}$, where $\beta_c$ denotes the inverse temperature, and $N$ denotes the number of responses in the given environment. We set $\beta_c = 3.0$ in all simulations. Then, the agent updates the Q-matrix according to the response and the reward. In all simulations, we employ fixed ratio (FR) 1 for other responses, where the agent can obtain rewards every time it engages in the responses and the reward values are 0.001 for all other responses. These flows are the same in all simulations. The only difference between the simulations is the schedule in which the environment gives rewards to the agents. Supplementary Algorithm 1 describes general flow of all simulations. In the following sections, we explain the differences in the schedules for each simulation.

*VR VI comparison and amount of training*. The VR schedule presents rewards depending on the response of the agent. At each response, the reward is presented at a given probability, which is the same as in the simulations. This means that reward presentation follows the Bernoulli process, and the number of responses required to obtain rewards follows the geometric distribution. We generate pseudo-random numbers following the distribution in order for the numbers to converge to the distribution in all simulations. More specifically, we divided the interval ranging from 0 to 1 into equal divisions according to the number of rewards, and the percentile points of the distribution were calculated for each point. Supplementary Algorithm 2 shows how to generate the required number of responses that follow the geometric distribution. We employ VR 15 in simulation 2.

The VI schedule presents rewards depending on the time lapse. However, the agent must emit responses to obtain rewards. Reward availability is determined at each time step according to a probability, and once the reward becomes available, it remains available until the agent takes the response. Reward availability follows the Poisson process, and the intervals between each reward follow an exponential distribution. Pseudo-random numbers are generated following the distribution in the same manner as the VR schedule. Supplementary Algorithm 3 shows how to generate inter-reward intervals that follow an exponential distribution. Moreover, we examined the effect of the amount of training on habit formation by manipulating the number of rewards in both schedules. We calculated the average of inter-reward intervals in the VR schedule and used them as the parameter of VI schedule.

*Comparison between choice and single schedule*. To examine the degree of habit formation when an explicit alternative is given, we used an environment that mimics the experiment conducted by Kosaki and Dickinson[15], where the effect of the presence or absence of the alternative on habit formation. Here, the agent can engage in two operant responses, and different rewards are assigned to each response. For example, two levers were inserted into the apparatus and pressing the left lever produced food, and the right levers produced a sucrose solution. In addition, as a control condition, we used an environment in which the agent can engage only one operant response, but the reward unavailable from the operant response is presented independent of the agent responses.

We mimicked these experiments. In the choice condition, two of the responses were treated as operant responses, and assigned two VI schedules with the same value and the reward values obtained from both were set to 1.0. In the no-choice condition, the operant response was assumed to be one, but the reward was presented independently of the response in order to control the reward amount. We assigned a variable time schedule to the rewards that are presented independent from the agent responses. We employ concurrent VI 60 VI 60 in the choice condition, and concurrent VI 60 VT 60 in the no choice condition. Supplementary Algorithm 4 and 5 describe the implementation of both of concurrent VI VI and VI VT schedules respectively.

*Tandem VI VR and tandem VR VI*. The tandem schedule is a schedule that presents multiple schedules in temporal succession. For example, tandem FR 5 VI 30 means that VI 30 will start after the agent has responded 5 times, and the reward will be presented at the end of VI 30. In addition, since tandem does not provide any explicit cues about the components it consists of, the agent can not know which schedule it is under. In tandem VI VR, the agent is first placed under a VI schedule, and after it is finished, it is moved to a VR schedule. In tandem VR VI the order of components are reversed, starting with the VI schedule and followed by

the VR schedule. We employ tandem VI 15 VR 3 and VR 10 VI 5. Supplementary Algorithm 6 and 7 describe the implementation of both of tandem VR VI and VR VI schedules respectively.

**Baseline and devaluation**. The reward devaluation is a procedure to examine whether an operant response is goal-directed or habit under free-operant situations. First, an animal learns that he or she can obtain a reward, food, or sucrose solution by pressing the lever. Learning lever pressings, the animal was placed in an experimental environment and trained to the operant response. After the training, reward devaluation was done by poisoning it with lithium chloride and a brief period was added where the animal can access the reward freely. Then, the animal was put into the experimental environment again and examined whether the number of operant responses decreased. If the number of responses does not change, it implies that the response is no longer controlled by its consequence, and the response becomes a habit. In contrast, if the number of responses decreases, the response is controlled by its consequences, such as goal-directed behavior. In our simulation, to reproduce the procedure, we reduced the value of the reward obtained from the operant response after the baseline phase.

*Baseline*. In the baseline phase, an agent travels on a network by choosing a response following Eq. (4) and searching for the shortest path between a currently engaging response and the goal. The simulation contains three steps: (1) choice of response based on reward values, (2) searching for the shortest path between the current response and the goal, and (3) engaging responses successively contained in the path. We calculated the proportion of an operant response to the total number of responses after some loops of the above 3 three steps. Supplementary Algorithm 1 shows the pseudocode of the simulation in the baseline phase.

*Devaluation*. In the devaluation phase, the agent behaved in the same way as in the baseline phase. The difference between the devaluation and baseline phases is only the value of the reward obtained from the operant response. In the baseline phase, we set the value to 1, and in the test phase, we set it to 0. At the baseline phase, we calculated the proportion of the number of operant responses to the total number of responses. Supplementary Algorithm 8 describes the procedure of the baseline and devaluation phase.

**Reporting summary**. Further information on research design is available in the Nature Portfolio Reporting Summary linked to this article.

## Data availability
All relevant data are within the paper (Figs. 2–8 and all Supplementary Figures) and the data and figures were generated using author's scripts (See Code availability).

## Code availability
All Python scripts written for the simulations and analysis are available at https://github.com/7cm-diameter/hbtnet.

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

## Acknowledgements

This research was supported by JSPS KAKENHI 20J21568 (KY), 18KK0070(KT), 19H05316 (KT), 19K03385 (KT), 19H01769 (KT), 22H01105 (KT), Keio Academic Development Fund (KT), Keio Gijuku Fukuzawa Memorial Fund for the Advancement of Education and Research (KT).

## Author contributions

Y.K. designed and performed simulations, analyzed data, and wrote the manuscript. K.T. supervised the study and edited the manuscript.

## Competing interests

The authors declare no competing interests.
