## [Peer Review File · Communications Biology]

Reviewers' comments:

Reviewer #1 (Remarks to the Author):

This paper proposes a network-based theoretical framework that tries to explain both the goal-directed mode and habitual mode of behaviors. Each node in the network represents a certain behavior and the strength of an edge corresponds to traditional Q-values in the learning literature. To show the validity of their model, they did two simulations. The first one is under given Q-values. And they claim that edge concentration corresponds to habit. In the second simulation, they try to link the known different factors in habit literature to the framework they proposed and show how the result changes under them.

Overall, I found this topic interesting and some novelty towards the direction they were trying. But unfortunately, I think the author did not convince me of their proposed theory for the following reasons.

The first thing that quite bothers me is that there is no state variable in the model they are proposing. Recall that most of the habit literature needs a state there (there are many there, Karamati 2016 PNAS will be one for instance). Whether distinguishing the state an agent is at is the key of differentiate these two modes. In the author's setup, there is no state there. In their Q-learning application, seems like the past action takes the role of the state. But it's quite different from the main habit theory literature. The authors should at least mention this set of literature and explain why they set things up differently, or seemingly different but the same. By the way, the authors should definitely cite Karamati 2016 PNAS as this is a paper about the hybrid mode of habit and goal-directed. I don't know why they are saying the rest of the field is treating these two modes completely separately.

And also, as a theory paper, I found both the mechanism and the process they are describing not clear enough. They say it's in the methodology session but I read the methodology session and still get the same confusion. With that being said, the way they are delivering their theory is probably too narrow and will probably not fit the interest of a general audience. The math is not that hard on this manuscript but the way they explain it is. I will give several specific examples of why I said that.

For example, what is the goal of this agent as we are in a multi stage game? Is he maximizing his total sum of values over all possible stage or what? The authors mention there are two goals, finding the max rewarded action and through the shortest path. But I don't understand why this is the goal of an animal or an agent? It's not a maze problem...The author should explain why this is the goal in a network full of actions. If it is this mouse is searching for cheese or a traveler is searching for treasures I got it. But for a sequence of behaviors, I don't quite get it. They may have a clear reason, but they just need to explain it well and put it clear in the main text. Again, it's still a set-up problem that the set up is explained well to the audience.

The second simulation with VI VR conditions seem interesting. But still, more explanations are needed. The first time VI VR appears in the main context, they are not even in full names! Readers will have no idea of why this is related to habit in the main text. The link from this model results and to our understanding of habit is just too vague. How should understand the cohesion of edges in a general way? What are the intuitions behind these simulation results?

Minor thing: equation does not have numbers in the pdf referees see.
Some of the math still lives in two dollar signs....

Overall, I feel this article may be promising but just needs to be polished a lot so that readers can understand what it conveys and what in addition we learned from it besides what's already known in the current literature.

Reviewer #2 (Remarks to the Author):

This study proposes a model of a network of behaviors that explains the formation of Habit. As far as I know, the model is new, and the idea of representing a chain of behaviors as a network is interesting. However, my concern is that the assumptions of the model seem to be unnatural, and its validity has not been sufficiently discussed. Therefore, it is difficult to evaluate the implications of the results at present. I think the authors should at least discuss the following points.

The authors assumed that the animal selects a target response, searches for the shortest path to that response, and then emits the sequence of responses according to the nodes along that path. However, the assumption that the animal is performing complex calculations such as finding the shortest path in a Dijkstra-like manner is somehow questionable.

The authors also state that they employed Q-learning to determine the network structure (although what they used is different from ordinary Q-learning in that it assumes Q-values for transitions from response to response, rather than for a response itself.) Although the Q-value is used only for network formation, the assumption that this Q-value itself is not directly used for action selection seems unnatural from the standard reinforcement learning perspective. If one assumes that the behavior is switched with probability depending on the strength of the Q-value, the model will likely produce the fixed sequence of behaviors even after the devaluation operation. Such a mechanism seems both sufficient and more parsimonious to explain habit formation. Why do we need to assume a goal-directed computation to find the shortest path in the network? If there is an advantage to such a computation, I would like the authors to discuss it.

In the update equation of Q-learning used by the author (line 158), it seems that only the immediate reward $R(a_t)$ was considered, but it is standard in Q-learning to consider the expected value of the cumulative reward from subsequent actions. The authors stated that the results were the same for SARSA (Line 303), but Q-learning and SARSA should be exactly the same algorithm if they do not take into account the reward from one or more future actions. They may be different in the authors' formulation, but at least they should be explicit about what kind of algorithm "SARSA" is that they used.

In addition, if the authors use an algorithm that takes into account the cumulative future rewards, the model may be able to form an interesting network structure such as a scale-free network, rather than a simple network structure in which only one operant response is a hub, as in the results of this study. However, it is somewhat questionable whether there is enough variation in animal responses to warrant consideration of network structures.

Minor points:

There are typos due to conversion errors around mathematical expressions. For example, the equation of Line 178 is " iin ", where " iin " is not converted to a symbol; Lines 193 and 219 are also not converted to equations.

Line 120 "We applicateintroduce..."

Reviewer #3 (Remarks to the Author):

In this paper Yamada and Toda present a theory of habit formation by representing the behavior as a

network of interconnected responses. Animals choose responses based on their rewards and then search for the shortest paths in the network to select actions. The authors further show that concentrated edges on the operant response leads to habit-like behavior, such as insensitivity to outcome devaluation. Other properties of the model are also investigated by studying different training conditions (such as duration, choice/no-choice, and training schedules).

In terms of presentation, the paper is overall well-written and the arguments are rather clear. It would benefit from a figure/graph which shows the process of response selection, shortest path search to help readers better understand the overall method. The model itself is interesting, but seems rather incomplete and missing some important elements of decision-making, as detailed below. As such, although I can see the novelty of the method in using network formulations, I think it needs further development in its current form to be ready for publication.

- The overall method is similar to a goal-directed planning (as the authors also mention), but there is no information about the 'state'; i.e., only a relationship between different responses and their rewards are learned by the animals in this model, and nothing about the relationship between actions and identity of their outcomes/states. I find it rather limiting since, firstly an important element of goal-directed learning is learning consequences of actions (beyond just their reward) and secondly, the connection to outcome-devaluation experiments is rather unclear -- as explained below:

- In the devaluation phase of outcome devaluation experiments, animals only "consume" the outcome without taking the operant action; this can be well accounted in model-based accounts of goal-directed behavior since outcomes are treated as states and relationship between actions and their consequences are learned in that model; however, in the current model, it is unclear how the reward of the operant response is updated in the devaluation phase, -- given that the operant response is not actually taken during the devaluation phase.

- The model is very similar in essence to the response-chaining/action-chucking kind of models. The main difference I could see is here responses are selected using the shortest path method, but in action chunking responses are selected in a chain. Based on this, it would firstly be important to better motivate the use of shortest path method, and secondly, to elaborate in what way it is different from action chunking theories.

- One of the features of habits is they are often considered automatic and fast (e.g., in terms of computational cost). Here it is rather unclear how transition from goal-directed actions to habits makes them faster. Indeed it seems the cost of shortest path search will make habits relatively costly and not necessarily cheaper than goal-directed actions.

- Based on the Figure 2d, it seems that the path to execute any response passed through an operant response. Does it imply that any response will be followed by an operant response? (i.e., grooming etc) -- is there any evidence to support this?

- There are some related works that aim to provide a single framework for both habits and goal-directed, actions, for example:

Pezzulo, Giovanni, Francesco Rigoli, and Fabian Chersi. "The mixed instrumental controller: using value of information to combine habitual choice and mental simulation." *Frontiers in psychology* 4 (2013): 92.

The authors can consider revising the manuscript to reflect this (e.g., "there is no theoretical framework that implements goal-directed behavior and 26 habits within a single system" in abstract).

Reviewer #1

General comment 1

This paper proposes a network-based theoretical framework that tries to explain both the goal-directed mode and habitual mode of behaviors. Each node in the network represents a certain behavior and the strength of an edge corresponds to traditional Q-values in the learning literature. To show the validity of their model, they did two simulations. The first one is under given Q-values. And they claim that edge concentration corresponds to habit. In the second simulation, they try to link the known different factors in habit literature to the framework they proposed and show how the result changes under them.

Overall, I found this topic interesting and some novelty towards the direction they were trying. But unfortunately, I think the author did not convince me of their proposed theory for the following reasons.

General reply

We greatly appreciate insightful and constructive comments on our manuscript. We have revised the manuscript thoroughly based on your comments and believe that the quality is much improved.

As you pointed out, we acknowledge that our manuscript was not kind for general readers. We mainly revised the following three points based on your comments.

1. We added an intuitive explanation of the model and schematic representation of that in the Figure 1 for helping understanding of general readers.
2. We described the previous studies simulated in our study in detail to clarify the relationship between those studies and our simulation.
3. We detailed our model to clarify that our model seems to have no state variable to avoid misleading.

These modifications made the manuscript more readable for general readers. We believe that the contributions of each simulation experiment to our theory became clearer in this revised manuscript.

Major comment 1 - 1

The first thing that quite bothers me is that there is no state variable in the model they are proposing. Recall that most of the habit literature needs a state there (there are many there, Karamati 2016 PNAS will be one for instance). Whether distinguishing the state an agent is at is the key of differentiate these two modes. In the author's setup, there is no state there. In their Q-learning application, seems like the past action takes the role of the state. But it's quite different from the main habit theory literature. The authors should at least mention this set of literature and explain why they set things up differently, or seemingly different but the same.

Reply to 1 - 1

Thank you for pointing out that the treatment of state variables in our model seems to differ from previous models and was not explained about it well. In our model, an immediate previous response is treated as a state and, in this sense, our model has a state variable. However, this formulation is different from previous ones. It comes from the difference in the experimental situation. Many previous models were proposed in the context of a multistate Markov decision process where a state is represented explicitly by choice points. Our model tried to explain habits in free-operant situations where no explicit state represents experimentally. In the case of free-operant situations, Perez and Dickinson (2020) assumed no state in their model and Daw et al. (2005) introduced a hypothetical state. Our approach, the immediate previous response is treated as the state, is similar to the later one.

We added sentences to Results and Discussion to clarify the treatment of the state variable in our model.

We stressed that a previous response is treated as a state by modifying Second paragraph of Result (L198 - 204),

We assumed that how nodes in a network and attachment of an edge between two nodes depended on the history of past rewards experienced by the agent. We employed Q-learning⁵⁵ to represent the history of rewards obtained when transitioning from one response to another. In ordinary Q-learning, an agent learns the action-value in a state. However, since our model dealt with transitions between responses, we treated the response of the agent as a state. Thus, Q-learning in our model was represented by the following equation, assigning the response a time point prior to the state:

We stressed that the treatment of state variable in our model is not so far from previous model by adding the following sentences to Second paragraph in Discussion (L623 - 634),

Second, our model seems to have no state variable, unlike previous models^{3,4,7,53}. We treated the immediately prior response of the agent as a state; thus, so there is no lack of state variables. This treatment of past responses as states has often been employed in modeling animal behavior^{32,33,51,54}. However, our model differs from past models of habits. Many models of habits were built in consideration of the multistage Markov decision task^{2,4,7}. In the multistage Markov decision task, experimentally explicit states, each choice point, exist. In contrast, we studied habits in free-operant situations in which animals could engage in responses freely and repeatedly, and experimentally explicit states were lacking. Previous models were applied to the free-operant situation in two different ways. One way was to not assume the state (1), and the other way was to introduce a hypothetical state^{7,53}. We treated the immediately prior response as a state, similar to the later one. Although our model seems to have no state variable, our approach was similar to the previous one^{7,53}.

Major comment 1 - 2

By the way, the authors should definitely cite Karamati 2016 PNAS as this is a paper about the hybrid mode of habit and goal-directed. I don't know why they are saying the rest of the field is treating these two modes completely separately.

Reply to 1 - 2

Thank you for pointing out our misunderstanding and introducing an important paper about it. The model employs a planning based on the reward history of an agent, and has common assumptions with our model. By citing the paper, we could improve our statement in the manuscript. We modified several sentences about this point and cited the paper in the main text.

We modified a sentence and added citation to refer existing hybrid mode model in the second paragraph of Introduction (L71 - 73),

However, some models explain habits in a multistage Markov decision task and challenge the canonical dichotomy of goal-directed and habits systems^{3,4}. In addition, some researchers reviewed existing studies on habit formation and cast doubt on the canonical framework of habit formation by showing the possibilities that habits are also controlled by their consequences^{5,6}.

To state the differences and similarities between our model and existing model (Keramati et al., 2016), we added a new paragraph as the following, in the third paragraph in the Introduction section (L85 - 92),

Another approach employs the planning process^{3,4}. Pezzulo et al.³ stressed the importance of planning in goal-directed behaviors and built a single mixed-controller model consisting of goal-directed behaviors and habits. Keramati et al.⁴ proposed that the canonical goal-directed and habits systems can be viewed as edges of the spectrum by building an integrated model of goal-directed planning and habits. Although application of their models was limited to the multistage choice task, the model could serve as a basis for a novel model with common assumptions and additional applicability in experiments on reward sensitivity in free situations¹¹⁻¹⁵.

Add the following sentences, stating that our model have common with existing model (Pezullo et al., 2013; Keramati et al., 2016) that were proposed in the context of the multi-stage Markov decision task and we showed their idea could be applied to the free-operant situations, in the second paragraph of the Discussion (L634 - 641),

Third, some models of habits assumed two distinct systems corresponding to goal-directed behavior and habits^{1, 2, 53}. Particularly, only the model that could explain habits in free-operant situations assumed them explicitly (1). Although all responses were assumed to be under goal-directed control, choices were based on reward values and shortest path search, and results reported in free-operant situations were reproduced¹¹⁻¹⁵. Recently, in the context of the multistage Markov decision task, several models showed no distinct systems between goal-directed behavior and habits^{3, 4}. Our model also showed no explicit distinction but that the idea could be applied to habits in free-operant situations.

Major comment 1 - 3

And also, as a theory paper, I found both the mechanism and the process they are describing not clear enough. They say it's in the methodology session but I read the methodology session and still get the same confusion. With that being said, the way they are delivering their theory is probably too narrow and will probably not fit the interest of a general audience. The math is not that hard on this manuscript but the way they explain it is. I will give several specific examples of why I said that.

For example, what is the goal of this agent as we are in a multi stage game? Is he maximizing his total sum of values over all possible stage or what? The authors mention there are two goals, finding the max rewarded action and through the shortest path. But I don't understand why this is the goal of an animal or an agent? It's not a maze problem....The author should explain why this is the goal in a network full of actions. If it is this mouse is searching for cheese or a traveler is searching for treasures I got it. But for a sequence of behaviors, I don't quite get it. They may have a clear reason, but they just need to explain it well and put it clear in the main text. Again, it's still a set-up problem that the set up is explained well to the audience.

Reply to 1 - 3

Thank you for pointing out barriers for understanding our model. We largely modified the description of the model in the first paragraph of the Result section to explain the intuition of

the model and a new figure to help understand what the agent does in the simulations. These modifications may help understand the model easier for general readers.

We modified the first paragraph of the Result to explain the intuition of our model and also added a new figure, as Figure 1B, to show the flow of behavior of our model (L163 - 197).

We considered the behavior of an agent as a network consisting of different categories of responses (e.g., lever pressing, grooming, stretching, etc.). Each response was assumed to be a node, and the transition between responses was assumed to be an edge (Figure 1A). The purpose of our agent was the same as the normal reinforcement learning setting of reward maximization. To achieve it, the agent's behavior was modeled by choices based on the values of rewards and the shortest path from the currently engaging response to the chosen response. Although this modeling differed from the ordinary setting, it accounted for the behavior of organisms in the natural environment. Our model reflected three facts (Figure 1B). (1) Most organisms, including humans, engage in various responses in their lives. For example, a rat in a free-operant experiment presses a lever in one moment and grooms its hair or explores the experimental apparatus the next moment. (2) The responses are associated with different types of rewards. Lever pressing is associated with food presentation. Hair grooming is associated with removing disconformity. Exploring within the apparatus is associated with escaping from the apparatus. (3) When an animal shifts from the currently engaging response to another response, it may choose to reach the response via relatively fewer responses. For example, if a rat engages in sniffing (Figure 1B left) and then chooses to press a lever (Figure 1B center), two paths or response sequences are available: walking to the front of the lever and pressing the lever or walking to the front of the lever followed by grooming and then pressing the lever (Figure 1B center). Grooming requires additional time and is redundant for pressing the lever. Thus, the rat may choose the shortest path, i.e., walk to the front of the lever and press it (Figure 1B right). In a large behavioral space, random search increases the time required to reach the desired response and does not warrant reaching the desired response. In summary, the agent chooses one available response associated with different

rewards and reaches the chosen response by following the shortest path from the currently engaging response. The agent loops through this process in the behavioral network, which is composed of responses.

Figure 1. Scheme of the behavioral network

A. The schematic representation of the behavioral network model represents how agents learn the Q-values by interacting with the environment and generate a behavioral network based on these values. The behavioral network consists of multiple responses. B. The schematic representation of the model's behavior shows how the agents transit in the network. The left panel shows the initial state in which agents engage in a response. The center panel shows that agents

choose a goal and search for the shortest path. The right panel shows that agents transit from the initial response to the goal via the shortest path.

Major comment 1 - 4

The second simulation with VI VR conditions seem interesting. But still, more explanations are needed. The first time VI VR appears in the main context, they are not even in full names! Readers will have no idea of why this is related to habit in the main text. The link from this model results and to our understanding of habit is just too vague. How should understand the cohesion of edges in a general way? What are the intuitions behind these simulation results?

Reply to 1 - 4

As you pointed out, our description about the simulation with VI and VR schedules was not kind for general readers to understand the relationship between habit formation and the simulation.

We added detailed descriptions and an intuitive example about the variable ratio and interval schedules in the first paragraph of the Simulation 2 section (L339 - 350).

The second factor is the rule, called schedule of reinforcement, which determines the criteria for presentation of a reward for a response¹³. Habit formation does not occur when reward presentation is determined by the number of responses by the animal. In this environment, the presence/absence of a reward is determined with a certain probability each time the animal presses a lever, e.g., in the bandit task or slot machine use. Habit formation occurs when rewards are determined according to the time elapsed since the previous reward. In this environment, the availability of a reward is determined potentially at arbitrary time steps with a certain probability, and the reward is presented at the first response after reward presentation becomes possible, such as checking a mailbox. The former response-based rule is called the variable ratio (VR) schedule, and the latter time-based rule is called the variable interval (VI) schedule. The third factor is the presence of alternatives.

We modified the third paragraph of the Interim Discussion section of the Simulation 2 to make it kind for general readers by adding a plain description of terms (VR and VI schedules) and an intuitive explanation why habit formation was promoted in VI schedule (L417 - 426).

The resistance to devaluation was larger in the VI schedule than in the VR schedule, suggesting that habit formation was promoted in the VI schedule. The VR schedule is a response-based rule of reward presentation. Therefore, all operant responses, independent of the agent's engagement immediately before, were rewarded with constant probability. In contrast, the VI schedule is a time-based rule and it causes that an operant response, longer elapsed time from last operant response, is selectively rewarded. In other words, an operant response emitted after a few periods was selectively rewarded and implied a transition from the other responses to the operant response in our model. In summary, transitioning from other responses to the operant response was selectively rewarded in the VI schedule and resulted in edge concentration in the operant response and habit formation.

Minor comment 1 - 1

Minor thing: equation does not have numbers in the pdf referees see.

Some of the math still lives in two dollar signs....

Reply to m1-1

Thank you for pointing out our errors and we added equation numbers and rendered all equations.

Reviewer #2

General comment 2

This study proposes a model of a network of behaviors that explains the formation of Habit. As far as I know, the model is new, and the idea of representing a chain of behaviors as a network is interesting. However, my concern is that the assumptions of the model seem to be unnatural, and its validity has not been sufficiently discussed. Therefore, it is difficult to evaluate the implications of the results at present. I think the authors should at least discuss the following points.

General reply

We greatly appreciate your helpful and insightful comments. We revised our manuscript extensively based on your comments. Now, we believe the quality of the manuscript is greatly improved in this revised manuscript.

In particular, the following three points have been modified significantly based on your comments in this revision.

1. We tried to replicate results of the simulation with looser constraints than shortest path search to generate response sequences.
2. In the original manuscript, we did not consider future rewards in Q-learning and we modified it to consider future rewards in this version.
3. We did not use Q-values to choose responses in the original version and we modified it to use them in this version.

In the revised manuscript, we significantly improved the model and re-conduct all simulations and showed that our original results are reproducible with the above changes. We also added some additional data to support these results.

We believe that these changes have corrected the flaws in the model that you have pointed out and have made both the model and manuscript significantly more convincing.

Major comment 2 - 1

The authors assumed that the animal selects a target response, searches for the shortest path to that response, and then emits the sequence of responses according to the nodes along that path. However, the assumption that the animal is performing complex calculations such as finding the shortest path in a Dijkstra-like manner is somehow questionable.

Reply to 2 - 1

We admit that a shortest path search, such as Dijkstra's algorithm, is not a realistic one that real animals do in nature. We employed the algorithm to ensure that an agent can find a path between one response to another because random search does not ensure that in a large behavioral space. We, therefore, claim that other algorithms can be applied if it can ensure that the agent can find the path. To prove our claim, we added a new simulation with another path finding algorithm which is more weakly constrained and not the shortest path searching algorithm.

We added a simulation and its result in the Supplementary figure 3. The detailed description of the algorithm is in the caption of the figure (L1197 - 1209).

Supplementary figure 3. Reproducibility of the results of Simulation 1 with a different response sequence generation algorithm.

In the Simulation 1, response sequences were generated by a shortest path search, Dijkstra's algorithm. We employed another algorithm that is more weakly constrained and not the shortest path searching algorithm. In the new algorithm, an agent chooses a response randomly if a response chosen as a goal is not connected to the current engaging response. If the goal response is connected to the current engaging response, the agent chooses the response. In other words, the agent searches the goal response locally in the new algorithm. Resistance to devaluation, Edge concentration and betweenness centrality, all of features are replicated with the new algorithm, suggesting habit formation does not depend on the shortest path search as long as the response sequences are generated goal-directed. Values in each line plots are mean \pm SEM (N = 10 for each point).

Major comment 2 - 2

The authors also state that they employed Q-learning to determine the network structure (although what they used is different from ordinary Q-learning in that it assumes Q-values for transitions from response to response, rather than for a response itself.) Although the Q-value is used only for network formation, the assumption that this Q-value itself is not directly used for action selection seems unnatural from the standard reinforcement learning perspective. If one assumes that the behavior is switched with probability depending on the strength of the Q-value, the model will likely produce the fixed sequence of behaviors even after the devaluation operation. Such a mechanism seems both sufficient and more parsimonious to explain habit formation. Why do we need to assume a goal-directed computation to find the shortest path in the network? If there is an advantage to such a computation, I would like the authors to discuss it.

Reply to 2 - 2

In the original version of our model, we use Q-values for generating the network. In the standard Q-learning model, Q-values are used for response selection. We modified it to use Q-values to choose responses in the Training phase. We re-conduct the Simulation 2 and 3 with the model and verify the results are replicated. However, we argue that goal-directed computation is still necessary. In a choice situation, as in the Simulation 2, the operant response remains goal-directed behavior even after extensive training. As shown in Figure 4, two operant responses acquired most of the edges in the network. Without goal-directed computation, both of the responses are chosen with almost the same probability, suggesting that habit formation occurs in the choice situation. Although both responses have almost the same number of edges, habit formation does occur. It implies that goal-directed computation is necessary to explain the result in the choice situation. Thus, the discussion of goal-directed computation in our model does not change from the original manuscript (L435 - 445).

We modified our model and added the following sentences in the second paragraph of the Learn Q matrix in the given Environment in Materials and Methods sections (L871 - 873).

The agent chooses a response according to the softmax function; $p_{i} = \frac{e^{\beta_c Q_i}}{\sum_{j=1}^N e^{\beta_c Q_j}}$, where β_c denotes the inverse temperature, and N denotes the number of responses in the given environment. We set β_c as x in all simulations.

We re-conducted all simulations in Simulation 2 and 3 sections, and the results of the original version of the model were replaced by the results of the modified ones.

Major comment 2 - 3

In the update equation of Q-learning used by the author (line 158), it seems that only the immediate reward $R(a_t)$ was considered, but it is standard in Q-learning to consider the expected value of the cumulative reward from subsequent actions. The authors stated that the results were the same for SARSA (Line 303), but Q-learning and SARSA should be exactly the same algorithm if they do not take into account the reward from one or more future actions. They may be different in the authors' formulation, but at least they should be explicit about what kind of algorithm "SARSA" is that they used.

Reply to 2 - 3

Thank you for pointing out our mistake in comparison between Q-learning and SARSA. In the original version of our model, future rewards were ignored and reward prediction errors were computed by only immediate rewards. However, we considered future rewards in the SARSA version of the model. Such a comparison is not fair, we admit. Thus we modified the Q-learning version of the model to consider future rewards and reconduct all simulations in Simulation 2 and 3. In addition, by considering future rewards, differences between the Q-value of the operant response and those of others decrease, and we modified the way to calculate a probability that an edge is attached between any two nodes. We showed all results are replicated by the new model and the comparison between Q-learning and SARSA is fair. These modifications support our claim that our findings are not limited to the specific algorithm.

We modified the original model for computing reward prediction errors by the new one that considers future rewards in the second paragraph of the Result section. In this modification, a new parameter, γ , was added and added a description about the parameter (L210 - 211).

$$Q(a_{t-1}, a_t) \leftarrow Q(a_{t-1}, a_t) + \alpha \cdot \delta \quad (1)$$

In this equation, γ denotes the discount rate of future rewards and we set $\gamma = 0.5$ for all simulations.

We modified the way to calculate a probability that an edge is attached between any two nodes. In the new model the probability is calculated by the softmax function with a new parameter. We modified the third paragraph of the Result section as the following (L214 - 222).

The probability that an edge is attached between any two nodes depends on the Q-value and is calculated by the softmax function. The probability was calculated using the following equation:

$$p_{i,j} = \frac{e^{-\beta_n Q(i,j)}}{\sum_{j=1}^N e^{-\beta_n Q(i,j)}} \quad (3)$$

In this equation, N denotes the number of nodes in the network and all the responses that the agent can engage in, and β_n is the inverse temperature, and we set $\beta_n = 50$ in all simulations.

Major comment 2 - 4

In addition, if the authors use an algorithm that takes into account the cumulative future rewards, the model may be able to form an interesting network structure such as a scale-free network, rather than a simple network structure in which only one operant response is a hub, as in the results of this study. However, it is somewhat questionable whether there is enough variation in animal responses to warrant consideration of network structures.

Reply to 2 - 4

Thank you for your profound and insightful comment. Although this paper does not exhaustively examine the various network structures, we believe that examining these different network structures will be beneficial in future research. As you pointed out, the network structure could be attributed to the scale-free network by considering future rewards. As I replied to your comment (Reply to 2 - 3), the differences between the Q-value of the operant response and those of others decreased by considering future rewards, implying such a possibility. In our result of the no choice situation in Simulation 2, the operant response acquired the most edges but several other responses also acquired not a few edges. It looks like the scale-free network, in precisely we must assess it by the distribution of degree, but habit formation occurred in the network. Although we did not compare scale-free networks with random network or hub-and-spoke networks, habit formation might be observed in the scale-free network. We revised our manuscript based on this discussion. In addition, as you are concerned, we do not know how many responses are included in the behavioral network in real animals. Although behavior estimation techniques allow researchers such analysis possible, it is not a well-established method at present. This technical limitation prevents us from providing a clear answer to your concern. However, as we show in Supplementary figure 2, our explanation of habit formation is applicable to networks of somewhat smaller size. Thus, we show that even if we did not have as many as 50 responses, we could still explain habit formation in the same way. We have revised the manuscript to clarify this discussion.

We added the following paragraph, in the last paragraph of the Interim Discussion of the Simulation 2, to discuss the relationship between our result and the scale-free network (L446 - 451).

In the no-choice situation, the operant response acquired the most edges in the network, but several other responses also acquired multiple edges (Figure 4B right), resembling the scale-free network, which should be assessed by the distribution of degree. However, habit formation occurred in the network. Therefore, although scale-free networks were not compared with random or hub-and-spoke networks, habit formation might be present in the scale-free-like network.

We added the following sentence about network size in the paragraph just before the Conclusion section (L812 - 826).

Recent advances in machine learning allow us to measure animal behavior more objectively and precisely than ever before. However, behavior estimation technologies are not well established at present, preventing us from validating some assumptions in our model. In this field, no consensus has been reached on what timescale should be employed to classify behavior and how finely behavior should be classified. For example, we assumed that the behavioral network consisted of 50 nodes but did not know how many nodes constitute the behavioral network of real animals. However, as shown in Supplementary Figure 2, habit formation occurred in networks of a slightly smaller size, suggesting that our explanation for habits could be applicable to the real behavioral network even if the size is smaller than we assumed. In the future, such technologies and by utilizing these techniques, it is possible to understand behavior on a macroscale rather than capture the behavior in highly constrained experimental settings. Our model provided a novel perspective on how behavior could be viewed on macroscale behavioral phenomena and raised questions that could be answered by such techniques, which would further help us understand the function of the brain in behavioral changes.

Minor comment 2 - 1

There are typos due to conversion errors around mathematical expressions. For example, the equation of Line 178 is "iin", where "in" is not converted to a symbol; Lines 193 and 219 are also not converted to equations.

Reply to m2-1

Thank you for pointing out our errors in equations and we fixed the equations and rendered all equations.

Minor comment 2 - 2

Line 120 "We applicateintroduce..."

Reply to m2-2

Thank you for pointing out an error in the main text and we deleted redundant characters.

Reviwer #3

General comment 3

In this paper Yamada and Toda present a theory of habit formation by representing the behavior as a network of interconnected responses. Animals choose responses based on their rewards and then search for the shortest paths in the network to select actions. The authors further show that concentrated edges on the operant response leads to habit-like behavior, such as insensitivity to outcome devaluation. Other properties of the model are also investigated by studying different training conditions (such as duration, choice/no-choice, and training schedules).

In terms of presentation, the paper is overall well-written and the arguments are rather clear. It would benefit from a figure/graph which shows the process of response selection, shortest

path search to help readers better understand the overall method. The model itself is interesting, but seems rather incomplete and missing some important elements of decision-making, as detailed below. As such, although I can see the novelty of the method in using network formulations, I think it needs further development in its current form to be ready for publication.

General reply

Thank you for reviewing our manuscript and giving constructive comments. We revised the manuscript based on your comments.

We revised the following x points in this revision.

1. We added a detailed explanation for the reason that our model seems to have no state variable.
2. We improve the description of the procedure to clarify how we simulate the reward devaluation procedure in this study.
3. We added a detailed relation between response chaining/action chunking models to clarify the similarity and difference between those models and ours.
4. We added new data to show that habit formation caused reducing computation cost in our model.

We revise problems you found and we believe that the manuscript is greatly improved.

Major comment 3 - 1

The overall method is similar to a goal-directed planning (as the authors also mention), but there is no information about the 'state'; i.e., only a relationship between different responses and their rewards are learned by the animals in this model, and nothing about the relationship between actions and identity of their outcomes/states. I find it rather limiting since, firstly an important element of goal-directed learning is learning consequences of actions (beyond just their reward) and secondly, the connection to outcome-devaluation experiments is rather unclear -- as explained below:

Reply to 3 - 1

In our model, the immediate previous response of an agent is treated as a state and our model has the state variable in this sense. If the agent chooses one response at a time step, it transits the response in the next time step and the response becomes the state. Thus, the agent need not learn state transition. However, such a modeling, the agent's response becoming the state, seems to differ from the standard model. It comes from the difference in the experimental situation. Many previous models were proposed in the context of a multistate Markov decision process where a state is represented explicitly by choice points. Our model tried to explain habits in free-operant situations where no explicit state represents experimentally. In the case of free-operant situations, Perez and Dickinson (2020) assumed no state in their model and Daw et al. (2005) introduced a hypothetical state. Our approach, the immediate previous response is treated as the state, is similar to the later one. Thus, our model is not so far from the existing model.

We added sentences to clarify that the previous response is treated as a state in our model in Second paragraph of Result (L198 - 204),

We assumed that how nodes in the network are connected, how an edge is attached between any two nodes, depends on the history of past rewards experience of the agent. We employed Q-learning (Watkins and Dayan, 1992) to represent the history of rewards obtained when transitioning from one response to another. In ordinary Q-learning, an agent learns action-value in a given state but since our model deals with transitions between responses, we treat the response of the agent as a state. Thus, Q-learning in our model is represented by the following equation, which assigns the response one time point prior to the state:

We added the following sentences to discuss about difference in the treatment of state variables between our model and existing models in Second paragraph in Discussion (L622 - 634),

Second, our model seems to have no state variable although previous models have them (Daw et al., 2005; Dezfouli and Balleine, 2012; Pezzulo et al., 2013; Keramati et al., 2016). In fact, we treat the immediate previous response of the

agent as a state, so there is no lack of state variables. This treatment of past responses as states has often been employed in modeling animal behavior (Killeen and Fatterman, 1988; Shull, 2011; Yamada and Kanemura, 2020; Sanabria et al., 2019). However, this formulation differs from past models for habits. Many of the models for habits are built with the multistage Markov decision task in mind (Daw et al., 2011; Dezfouli and Balleine, 2012; Pezzulo et al., 2013; Keramati et al., 2016). In the multistage Markov decision task, experimentally explicit states, each choice point, exist. In contrast, we tried to explain habits in free-operant situations where animals can engage any response freely and repeatedly, and there are no experimentally explicit states. Previous studies attempted to apply the models to the free-operant situation in two different ways. One way is to not assume the state (Perez and Dickinson, 2020), and another is to introduce a hypothetical state (Daw et al., 2005; Dezfouli and Balleine, 2012). We treat the immediate previous response as a state and it is similar to the later one. Although our model seems to have no state variable, our approach is not so far from the previous one (Daw et al., 2005; Dezfouli and Balleine, 2012).

Major comment 3 - 2

In the devaluation phase of outcome devaluation experiments, animals only “consume” the outcome without taking the operant action; this can be well accounted in model-based accounts of goal-directed behavior since outcomes are treated as states and relationship between actions and their consequences are learned in that model; however, in the current model, it is unclear how the reward of the operant response is updated in the devaluation phase, -- given that the operant response is not actually taken during the devaluation phase.

Reply to 3 - 2

Thank you for pointing out an unclear description of the reward devaluation procedure and we found misleading expressions in our manuscript.

In general, the reward devaluation procedure is conducted using the following steps. (1) Animals trained under an arbitrary environment, such as VR, VI, or choice situations. (2)

They experienced the reward devaluation by poisoned reward or satiation of the reward outside the experiment. (3) Animals are returned to the experimental environment and assessed whether their behavior has become habits or not. In the test, animals can engage the operant response freely but no reward is delivered at the time. It implies the animals have learned the value of the reward outside the experiment, we described in (2), but not in the test. We replicate these steps in our simulation. Since the reward devaluation is done outside of the experiment, our model does not learn the reward value within the simulation in the same way. Instead, we change the reward value obtained by the operant response from 1 to 0 when moving from the baseline to the test phases.

We revise the description of the reward devaluation procedure by modifying the first paragraph of the Simulation 1 as the following.

To examine the degree of habit, we used the reward devaluation procedure used in free-operant experimental situations. The earliest demonstrations of habit formation¹¹⁻¹³ used the reward devaluation procedure. In this procedure, the investigators train the animals to press the lever with a reward. After the animal learned lever pressings to obtain the reward, the value of reward was reduced by poisoning it with lithium chloride. In this procedure, **animals learnt the reward value outside the experiment. Subsequently, investigators examined if the animal pressed the lever without reward deliveries, or an extinction test. Thus, the reward value for the animal was not updated in the test. When the animal pressed the lever, the reward was poisonous, and the responses were considered to be a habit. When the lever-presses decreased after devaluation, the responses were considered to be goal-directed behavior.** To reproduce the procedure in the simulation setting, we set up the baseline and devaluation phases where the value of reward obtained by the operant response is 1 and 0, respectively. **As animals had experienced reward devaluation outside the experiments in the experimental setting, our agents did not update the reward value within the simulation but changed it from 1.0 to 0.0 before starting when moving from baseline to test phases.**

Major comment 3 - 3

The model is very similar in essence to the response-chaining/action-chucking kind of models. The main difference I could see is here responses are selected using the shortest path method, but in action chunking responses are selected in a chain. Based on this, it would firstly be important to better motivate the use of shortest path method, and secondly, to elaborate in what way it is different from action chunking theories.

Reply to 3 - 3

First, the adoption of shortest path search does not play an essential role in our model. As the additional simulation and its results (Supplementary figure 3) show, our original results are reproduced if the reaction series is generated by an objective-oriented algorithm, even if not by shortest path search. This result also stresses the importance of goal-directed computation, such as planning, but suggests that its implementation could be more simplified. Second, the biggest difference between the response-chaining/action-chucking models and our model is targeting experimental situations and the way of viewing behavior. Many of the former models were built with multistage Markov decision tasks in mind. Our model, on the other hand, was built to explain habit formation in free operant situations. Actually, response-chaining/action-chucking models were not applied to free-operant situations comprehensively. We considered the behavior as a network and applied the kind of idea like chain or chunk into free operant situations.

Discussion and additional simulations were made regarding the adoption of the shortest path. As supplementary figure 3, we added the following figure (L1197 - 1209).

Reproducibility of simulation results with a different goal selection strategy in Simulation 1

Supplementary figure 3. Reproducibility of the results of Simulation 1 with a different response sequence generation algorithm.

In the Simulation 1, response sequences were generated by a shortest path search, Dijkstra's algorithm. We employed another algorithm that is more weakly constrained and not the shortest path searching algorithm. In the new algorithm, an agent chooses a response randomly if a response chosen as a goal is not connected to the current engaging response. If the goal response is connected to the current engaging response, the agent chooses the response. In other words, the agent searches the goal response locally in the new algorithm. Resistance to devaluation, Edge concentration and betweenness centrality, all of features are replicated with the new algorithm, suggesting habit formation does not depend on the shortest path search as long as the response sequences are generated goal-directed. Values in each line plots are mean \pm SEM (N = 10 for each point).

We modified the last two paragraphs in the relationship to other theoretical models of habits in the Discussion to make it clear the difference between our model and response-chaining/action-chunking models (L671 - 708).

However, the models have two differences. First, the targeting experimental situations differed. Their model was built with the multistage Markov decision task, while our model was built to explain habit formation in free-operant situations. The existing comprehensive theory in free-operant situations assumed parallel control by two systems (1). A kind of response-chaining/action-chunking models have limited applicability in free-operant situations. Second, the view of behavior differed. Our model tried to overcome the limitation. In free-operant situations, animals could engage in responses freely without explicit states defined experimentally. In the case of free-operant situations, direct application of the idea of response-chaining or action-chunking was difficult because no points corresponded to the start and end of trials. Instead of the chunk or chain, we considered behavior as a network and the agent's behavior as a transition within the network. In other words, by viewing behavior as a loop without a clear start or end, we successfully modeled the behavior of free-operant situations.

Dezfouli and Balline⁷ applied their model to the free operant situation and reproduced the effect of amount of training on habit formation. However, they did not treat how other factors, schedule types and presence of alternatives, affect habit formation. The proposed model, which shares common assumptions with their model, can reproduce the results reported in empirical study¹¹⁻¹⁵, suggesting that the idea of response-chaining or action-chunking could be applied in free-operant situations. Moreover, the model clarifies the difference between the canonical correlation-based account and common points with the contiguity-based account. We also found common features with the recently proposed models^{3, 4}. In those models, goal-directed planning was employed, and the behavior of human and rodents' multistage decision-making tasks, such as multistage Markov decision tasks and tree-shaped maze, were explained. Pezzulo et al.³ built a mixed-controller model consisting of goal-directed and habit behaviors in a single system. Keramati et al.⁴ proposed that these two systems were not separated but placed in one spectrum. Our model also considered these two systems to be not separated but coexisting in a single system and

placed in one spectrum, with only a difference in the structure of the network. However, similar to many other models, their models targeted multistage decision-making tasks but not free-operant situations. Our model shared common features, i.e., planning and singularity of the system, with their models³ and successfully applied those features in free-operant situations. From the canonical view, two distinct systems control a response in the flat manner^{1,2}. This view has been challenged recently, and new models have been proposed in the context of the multistage decision-making tasks. Although their applications are limited to free-operant situations, our model adopted those ideas, i.e., response-chaining/action-chunking, planning, and mono-systematicity, and explained habit formation in free-operant situations, suggesting a link between the different experimental procedures and providing a comprehensive understanding of habit formation.

Major comment 3 - 4

One of the features of habits is they are often considered automatic and fast (e.g., in terms of computational cost). Here it is rather unclear how transition from goal-directed actions to habits makes them faster. Indeed it seems the cost of shortest path search will make habits relatively costly and not necessarily cheaper than goal-directed actions.

Reply to 3 - 4

Thank you for finding an interesting point. We added a new result to show that computation cost decreased when habit formation occurred. In our model, habit formation occurred by the edge concentration to the operant response. When most edges are connected to the operant response, transition from one response to another can be achieved via the operant response. Thus, the agent can find the shortest path easily and make the transition faster.

We added a new figure as Figure 3 and it shows the average distance between two nodes and the required time for simulation. We also add the description of the result in the Simulation result in the Simulation 1 section (L281 - 284 and L299 - 307).

With edge concentration in the operant response, distances between two nodes in the network decreased (Figure 3 left). Furthermore, transitions made by agents in the simulations became efficient, and time required for simulations shortened (Figure 3 right).

Figure 3. Reduced computation costs with habit formation

The left panel shows the average path length, i.e., the average of the shortest path between two nodes in the network. When the path length is shorter, the transition from one response to another becomes faster. The right panel shows the required time to simulate the baseline phase. The required time is the real time, i.e., the duration from the start to the end of the simulation. Since the number of loops is the same for all simulations, the decrease in required time implies efficiency in shortest path search and transitions between responses. Values in each line plots are mean \pm SEM (N = 10 for each point).

We added a paragraph to discuss the result in the last paragraph of the Interim Discussion in the Simulation 1, as the following (L320 - 331).

Habits are efficient in the computational cost and transition^{7,38}. In our model, these features of habits were also found. Animal responses are constrained by some factors, such as space and the animal's body. For example, an animal cannot eat food if the food is not in front of it and if it cannot walk when it is sleeping.

These examples imply that not all responses are connected to each other and that the number of edges in the network is limited. When the number of edges was constrained, the structure of the network promoted that agent to engage in the desired response. When edges from other responses were concentrated in the operant response, the average distance between two nodes was shortened³⁹, and transitions made by agents became efficient (Figure 3). These results also imply that agents can find the path between two nodes faster. Thus, habit formation, i.e., edge concentration to the response, reduces the computational cost and hastens the transition under constraints.

Major comment 3 - 5

Based on the Figure 2d, it seems that the path to execute any response passed through an operant response. Does it imply that any response will be followed by an operant response? (i.e., grooming etc) -- is there any evidence to support this?

Reply to 3 - 5

At present, we have no clear answer for this. Although behavior estimation techniques enable researchers to conduct such analysis, it is not a well-established method at present. In particular, there is no consensus on what timescale to be employed to classify behavior and how finely behavior should be classified. We expect the point will be resolved by future technological developments and research progress. We considered that our contribution is rather that we have created questions that can be tested by large-scale behavioral measurements achieved by technological developments.

We modified the last paragraph in the Discussion section by reflecting your point as the following (L812 - 826).

Recent advances in machine learning allow us to measure animal behavior more objectively and precisely than ever before. **However, behavior estimation technologies are not well established at present, preventing us from validating**

some assumptions in our model. In this field, no consensus has been reached on what timescale should be employed to classify behavior and how finely behavior should be classified. For example, we assumed that the behavioral network consisted of 50 nodes but did not know how many nodes constitute the behavioral network of real animals. However, as shown in Supplementary Figure 2, habit formation occurred in networks of a slightly smaller size, suggesting that our explanation for habits could be applicable to the real behavioral network even if the size is smaller than we assumed. In the future, such technologies and by utilizing these techniques, it is possible to understand behavior on a macroscale rather than capture the behavior in highly constrained experimental settings. Our model provided a novel perspective on how behavior could be viewed on macroscale behavioral phenomena and raised questions that could be answered by such techniques, which would further help us understand the function of the brain in behavioral changes.

Major comment 3 - 6

There are some related works that aim to provide a single framework for both habits and goal-directed, actions, for example:

*Pezzulo, Giovanni, Francesco Rigoli, and Fabian Chersi. "The mixed instrumental controller: using value of information to combine habitual choice and mental simulation." *Frontiers in psychology* 4 (2013): 92.*

The authors can consider revising the manuscript to reflect this (e.g., "there is no theoretical framework that implements goal-directed behavior and 26 habits within a single system" in abstract).

Reply to 3 - 6

Thank you for pointing out our misunderstanding and introducing the important paper that tried to explain goal-directed behavior and habits in a single system and employed a planning process as our model does. By citing the paper and adding several sentences, our

manuscript has been improved by clarifying the relationship between the previous model and ours.

We modified the pointed sentence as the following (L22).

Although canonical theories based on such distinctions are starting to be challenged, **a few** theoretical frameworks that implement goal-directed behavior and habits within a single system.

We added the following sentence to address that the model tried to canonical dichotomy in the second paragraph of the Introduction section (L71 - 73).

However, some models explain habits in a multistage Markov decision task and challenge the canonical dichotomy of goal-directed and habits systems^{3,4}.

We added the following sentences to clarify the similarity and difference between our model and existing models in the third paragraph of the Introduction (L85 - 92).

Another approach employs the planning process^{3,4}. Pezzulo et al.³ stressed the importance of planning in goal-directed behaviors and built a single mixed-controller model consisting of goal-directed behaviors and habits. Keramati et al.⁴ proposed that the canonical goal-directed and habits systems can be viewed as edges of the spectrum by building an integrated model of goal-directed planning and habits. Although application of their models was limited to the multistage choice task, the model could serve as a basis for a novel model with common assumptions and additional applicability in experiments on reward sensitivity in free situations¹¹⁻¹⁵.

In the second paragraph of the Discussion, we added the following sentences to clarify that there have been models that tried to explain goal-directed behavior and habits in a single system. (L634 - 641).

Third, some models on habits assumed two distinctive systems corresponding to goal-directed behavior and habits (Daw et al., 2005; Daw et al., 2011; Perez and Dickinson; 2020). In particular, only the model that can explain the habits in the free-operant situations assume them explicitly (Perez and Dickinson; 2020). Although we assumed that all responses are under the goal-directed control, choices based on reward values and shortest path search, we reproduced the

results reported in free-operant situations (Adams, 1982; Dickinson, et al., 1995; Dickinson, Nicholas, and Adams, 1983; Colwill and Rescorla, 1985; Kosaki and Dickinson, 2010). Recently, in the context of the multistage Markov decision task, several models do not have explicit distinctive systems between goal-directed behavior and habits (Pezzulo et al., 2013; Keramati et al., 2016). Our model also does not have the explicit distinction, and showed that the idea can be applied to habits in the free-operant situations.

In the last paragraph of the Relationship between other theoretical models of habit formation in the Discussion section, we added sentences to clarify the similarity and difference between our model and existing model as the following (L691 - 708).

We also found common features with the recently proposed models^{3,4}. In those models, goal-directed planning was employed, and the behavior of human and rodents' multistage decision-making tasks, such as multistage Markov decision tasks and tree-shaped maze, were explained. Pezzulo et al.³ built a mixed-controller model consisting of goal-directed and habit behaviors in a single system. Keramati et al.⁴ proposed that these two systems were not separated but placed in one spectrum. Our model also considered these two systems to be not separated but coexisting in a single system and placed in one spectrum, with only a difference in the structure of the network. However, similar to many other models, their models targeted multistage decision-making tasks but not free-operant situations. Our model shared common features, i.e., planning and singularity of the system, with their models^{3,4} and successfully applied those features in free-operant situations. From the canonical view, two distinct systems control a response in the flat manner^{1,2}. This view has been challenged recently, and new models have been proposed in the context of the multistage decision-making tasks. Although their applications are limited to free-operant situations, our model adopted those ideas, i.e., response-chaining/action-chunking, planning, and mono-systematicity, and explained habit formation in free-operant situations, suggesting a link between the different experimental procedures and providing a comprehensive understanding of habit formation.

In the last paragraph of the Neural substrates of the behavioral network in the Discussion section, we added descriptions citing the suggested paper (L746 - 751).

Neuronal circuits involving ventral striatum and hippocampus play key roles in spatial navigation and are considered to be related to the planning^{72,73}. Both spatial navigation and planning are related to habits, and they share common neurobiological substrates^{3,74-78}. Although roles of hippocampus and planning in habits and goal-directed behavior in free-operant situations remains unknown, our model sheds light on the role of planning and related brain regions in habits in the free-operant situations.

Reviewers' comments:

Reviewer #2 (Remarks to the Author):

I believe the authors have made a sufficient effort to respond to my comments: The authors added new model elements and ran additional simulations. While I feel that there is future work to be done on the validity of this model, I feel that this paper is worthy of publication as it proposes a model that explains the mechanism of habit formation.

Reviewer #3 (Remarks to the Author):

I would like to thank the authors for their responses to my comments. Although the paper has improved (e.g., Figure 3 is a very nice addition), unfortunately, I feel the responses only partially address my major concerns. I will provide details below.

Major comment 3-2: As noted by myself and the authors, during the devaluation phase, animals are only exposed to outcomes but not responses, and therefore, the reward of the responses will/should remain unaffected by the devaluation experience. This is because the model only learns response rewards (but not the identity of their outcomes). For the devaluation to affect response reward, the model should learn the identity of the outcomes of each response, which is not the case in the proposed model.

Based on this, it is unclear how the response value will change from 1 to 0 after the devaluation (as mentioned by the authors). It is, in my view, not a minor detail, but it shows that the lack of state (outcome) information is a serious limitation of this model (as also mentioned by other reviewers) for explaining the devaluation effect.

Related to the above comment, the authors have mentioned that the previous action represents state information; but it is unclear how the model handles tasks like the above explained above, cues in the environment, or tasks with multiple outcomes (e.g., two actions each leading to separate outcomes and only one of the outcomes is devalued).

Indeed, contrary to what the authors have mentioned, free operant setup and state information are not mutually inconsistent. Previous models can work with free operant conditions and maintain state information.

I would like to thank the authors for their responses to my comments. Although the paper has improved (e.g., Figure 3 is a very nice addition), unfortunately, I feel the responses only partially address my major concerns. I will provide details below.

We thank you for your previous comments which have greatly improved our manuscript and also pointing out your concerns in detail. We have noticed that there may be a discrepancy in understanding of the procedure of simulations between you and us. We believe that your serious concerns could be substantially untangled by resolving this discrepancy. We have clearly explained this point and added a figure to explain the procedure in detail. We believe this revision worked out the miscommunication between you and us and improved the quality of the manuscript.

Major comment 3-2: As noted by myself and the authors, during the devaluation phase, animals are only exposed to outcomes but not responses, and therefore, the reward of the responses will/should remain unaffected by the devaluation experience. This is because the model only learns response rewards (but not the identity of their outcomes). For the devaluation to affect response reward, the model should learn the identity of the outcomes of each response, which is not the case in the proposed model.

We are very sorry that our description about the procedure of the simulation was confusing. Perhaps the step we call the devaluation phase and what the reviewer calls the devaluation phase refer to different things in the overall procedure. We believe that the reviewer's concerns can be addressed by revising the description of the procedures and resolving discrepancy between the authors and the reviewer. We have added a new figure. The figure clearly demonstrates the correspondence between the procedures of the animal experiment and each step of the simulation.

In this revision, we modified the description of the procedure of the simulation and added the schematic representation of the procedure. After agents constructed a network based on Q-matrix, they were exposed to the baseline phase where the reward value acquired by the operant response remains intact. After the baseline phase we changed the reward value (first element of r in equation #) of the operant response from 1 to 0. We suspect that the reviewer took this step as the devaluation phase. After the change in the reward value, agents were exposed to the same situation and we called this step the "devaluation phase". We found that the name of

steps in our simulation were confusing. By the reviewer's points, we realized that our naming is very confusing and we have modified them. Again, thank you so much for your comment.

We modified the first paragraph of the Overview in Materials and Methods section as followings (L834 - 854),

We conducted three simulations in this article, and they contain four steps (Figure 9). In the first step, agents learn the Q matrix in the given environments in the simulation 2 and 3 but agents are given a hypothetical Q matrix in the simulation 1. After the training phase, the agents generate a network based on the Q matrix. The way to generate the network is the same in all simulations. In the second step, the baseline phase, the agents travel in the network and engage responses. Here, the agents choose their responses based on reward values and the reward value obtained by the operant response is set to 1.0. The agents no longer update the Q matrix nor reconstruct the network. In the third step, the devaluation phase, the reward value of the operant response (in Eq. 4) was reduced from 1.0 to 0.0 without any interaction with environments. If there were two operant responses, the value of only one of them (reduced but not) was reduced. In the fourth step, the post devaluation phase, the agents behave in the same way as the baseline. However, the reward value of operant response is reduced to 0.0. The only difference between baseline and devaluation is the reward value of operant response. We explain the procedures conducted in the four steps in detail after sections. Our simulation codes are available at: <https://github.com/7cm-diameter/hbtnet>.

Figure 9. Overview of simulations.

The upper panel shows the schematic representation of the reward devaluation procedure in real experiments. The bottom panel shows the simulation procedure as corresponding to the empirical procedure.

Based on this, it is unclear how the response value will change from 1 to 0 after the devaluation (as mentioned by the authors). It is, in my view, not a minor detail, but it shows that the lack of state (outcome) information is a serious limitation of this model (as also mentioned by other reviewers) for explaining the devaluation effect.

We did not change the response value (if you meant action-value function) after agents learned it in the training phase. We only changed the reward value acquired by the operant response. We hope that the added Figure 9 will resolve this concern.

Related to the above comment, the authors have mentioned that the previous action represents state information; but it is unclear how the model handles tasks like the above explained above, cues in the environment, or tasks with multiple outcomes (e.g., two actions each leading to separate outcomes and only one of the outcomes is devalued).

By modified figures and descriptions, we expect this concern about simulation procedures is resolved now.

Indeed, contrary to what the authors have mentioned, free operant setup and state information are not mutually inconsistent. Previous models can work with free operant conditions and maintain state information.

We did not suggest that the state variable and free-operant situation is mutually inconsistent. We only suggested that the state is not always needed for a free-operating situation.

We added the following sentence to the second paragraph of the Discussion section (L624 - 626).

Thus, we suggest that the inclusion or exclusion of state variables to explain habit formation in free-operant situations depends on the details of the model and is not always necessary.